# DECOUPLED ACTOR-CRITIC

## ABSTRACT

Actor-critic methods are in a stalemate of two seemingly irreconcilable problems. Firstly, critic proneness towards overestimation requires sampling temporal-difference targets from a conservative policy optimized using lower bound Q-values. Secondly, well-known results show that policies that are optimistic in the face of uncertainty yield lower regret levels. To remedy this dichotomy, we propose Decoupled Actor-Critic (DAC). DAC is an off-policy algorithm that learns two distinct actors by gradient backpropagation: a conservative actor used for temporal-difference learning and an optimistic actor used for exploration. We test DAC on DeepMind Control tasks in low and high replay ratio regimes and ablate multiple design choices. Despite minimal computational overhead, DAC achieves state-of-the-art performance and sample efficiency on locomotion tasks.

## 1 INTRODUCTION

Deep Reinforcement Learning (RL) is still in its infancy, with a variety of tasks unsolved (Sutton & Barto, 2018; Hafner et al., 2023) or solved within an unsatisfactory amount of environment interactions (Zawalski et al., 2022; Schwarzer et al., 2023). Whereas increasing the replay ratio (ie. the number of parameter updates per environment interactions step) is a promising general approach for increasing sample efficiency and final performance of RL agents (Janner et al., 2019; Chen et al., 2020; Nikishin et al., 2022; Li et al., 2022), it is characterized by quickly diminishing gains (D'Oro et al., 2022) combined with linearly increasing computational cost (Rumelhart et al., 1986; Kingma & Ba, 2014). Moreover, the limitations of robot hardware and data acquisition frequency constrain the maximum achievable replay ratio (Smith et al., 2022). As such, it is worthwhile to pursue orthogonal techniques such as enhancing the properties of the underlying model-free agents. One continuously researched theme is how a particular algorithm handles the *exploration-exploitation* dilemma (Hessel et al., 2018; Fujimoto et al., 2019; Ciosek et al., 2019; Ecoffet et al., 2021).

In Actor-Critic (AC) algorithms, it's common to employ a single policy for both exploration (gathering new data to improve the current best policy) and exploitation (leveraging gathered data to determine the best policy) (Silver et al., 2014; Schulman et al., 2015; Wu et al., 2017; Schulman et al., 2017; Moskovitz et al., 2021). Algorithms like TD3 (Fujimoto et al., 2018) or SAC (Haarnoja et al., 2018) achieve exploration by introducing symmetric noise to an exploitative action. However, this noisy exploitation strategy necessitates careful balancing of policy entropy (Duan et al., 2016). Whereas insufficient entropy leads to suboptimal policies due to inadequate exploration (Haarnoja et al., 2018), excessive entropy results in suboptimal policies due to noisy critic network updates and, consequently, poor Q-value approximator convergence (Van Seijen et al., 2009). Additionally, optimizing the policy towards Q-value lower bound leads to an inadequate exploration of the state-action subspace that yields critic disagreement (Ciosek et al., 2019; Moskovitz et al., 2021).

Using a single policy for both exploration and exploitation in AC algorithms has its roots in the Policy Gradient (PG) Theorem (Sutton et al., 1999) which states that PG is a function of Q-values under the current policy. Thus, approaches building on PG would often use SARSA-type updates to train the critic (Silver et al., 2014; Hafner et al., 2019a; Cetin & Celiktutan, 2023), as SARSA converges to on-policy Q-values (Sutton & Barto, 2018). This in turn reinforces a single-policy setup for AC algorithms. Recently, there have been works in relaxing the PG Theorem toward a dual-policy, fully off-policy setup (Laroche & Tachet des Combes, 2021). An example of a dual-policy implementation is Optimistic Actor-Critic (OAC) (Ciosek et al., 2019). OAC uses two policies: optimistic for exploration (ie. sampling actions when interacting with the environment); and conservative for exploitation (ie. sampling actions for temporal-difference learning). Both policies are extracted

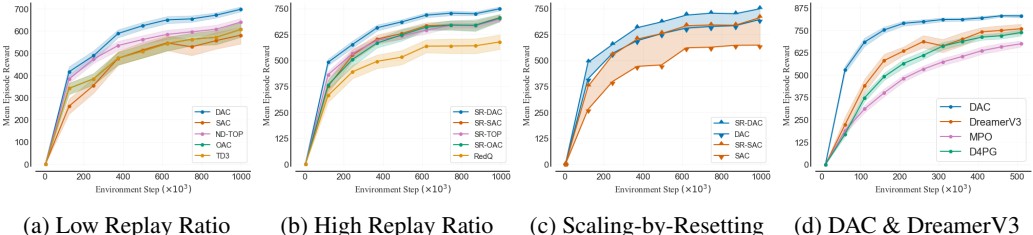

(a) Low Replay Ratio    (b) High Replay Ratio    (c) Scaling-by-Resetting    (d) DAC & DreamerV3

Figure 1: Decoupled Actor-Critic (DAC) achieves significant improvements on DeepMind Control Suite despite minimally higher computational costs than Soft Actor-Critic (SAC), demonstrating state-of-the-art performance in complex locomotion tasks. Figure 1a reports compute efficient experiments, where algorithms perform only 3 updates per environment step. Figure 1b reports a sample efficient, high replay ratio experimental setup. Figure 1c shows that DAC matches the performance of state-of-the-art Scaled-by-Resetting (SR) SR-SAC despite using 5-times lower replay ratio and no parameter resets, whereas SR-DAC outperforms both. Figure 1d compares low replay DAC, DreamerV3, MPO and D4PG. The Figure highlights DAC's competitive performance despite significantly lower complexity, runtime, and computational demands than the model-based algorithm. We detail the setting in Section 4 and Appendix G. 10 seeds, mean and 95% bootstrapped CI.

from a single conservative actor. Whereas the conservative policy is directly parameterized by the actor, the optimistic policy stems from a local linear approximation of Q-value upper bound constrained by a desired Kullback-Leibler (KL) divergence. This yields an approximation of a policy that is Optimistic in the Face of Uncertainty (OFU) (Wang et al., 2020b; Neu & Pike-Burke, 2020). Unfortunately, OAC exploration is highly dependent on the chosen hyperparameter values.

To address the above shortcomings, we propose Decoupled Actor-Critic (DAC). DAC tackles the exploration-exploitation dilemma by adopting a novel decoupled actor AC approach. As such, DAC employs two actors, each independently optimized using gradient backpropagation with different objectives. The optimistic actor is trained to maximize an optimistic Q-value upper bound while adjusting optimism levels automatically. This actor is responsible for exploration (sampling transitions added to the experience buffer). In contrast, the conservative actor is trained using standard lower bound soft policy learning (Haarnoja et al., 2018) and is used for sampling temporal-difference (TD) targets and evaluation. Secondly, DAC addresses the shortcomings of OAC. By relaxing the first-order Taylor approximation and explicitly modeling the second policy via an actor network DAC can accurately approximate the maximum of arbitrary complexity Q-value upper bound (Hornik et al., 1989). We perform experiments summarized in Figure 1, and show that DAC outperforms other model-based RL agents. We highlight the main contributions of DAC below:

- We propose a novel off-policy dual-actor AC setup where each actor is trained via gradient backpropagation of a specialized objective. We define the optimistic policy objective and formulate a robust framework that introduces easily interpretable hyperparameters.

- We implement a module that automatically adjusts the level of optimism applied during Q-value upper bound approximation, as well as the impact of the KL penalty. This in turn allows DAC to accommodate various levels of epistemic and aleatoric uncertainties and different reward scales without hyperparameter tuning.

- We show that DAC outperforms model-free benchmarks in terms of both sample efficiency and final performance, in both low and high replay regimes. To facilitate further research, we perform extensive ablations on various design choices (over 2000 training runs). We release training logs, as well as implementations of DAC under the following URL.

## 2 PRELIMINARIES

In this paper, we address policy learning in continuous action spaces. We consider an infinite-horizon Markov Decision Process (MDP) (Puterman, 2014) which is described with a tuple $(S, A, R, p, \gamma)$, where states $S$ and actions $A$ are continuous, $R(s, a, s')$ is the transition reward, $p(s'|s, a)$ is a transition kernel and $\gamma \in (0, 1]$ is a discount factor. A policy $\pi(a|s)$ is a state-conditioned action

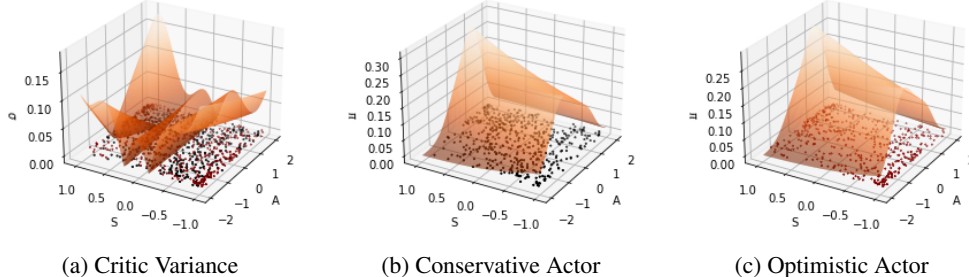

(a) Critic Variance       (b) Conservative Actor       (c) Optimistic Actor

Figure 2: Pessimistic underexploration and state-action space coverage on the Pendulum task with state representation embedded into 1 dimension. The dots represent 500 state-action samples gathered using the latest policy (conservative (black) or optimistic (red)). Figure 2a displays the standard deviation ($\sigma$) of the two critics, with smaller values observed in well-explored state-action regions. In Figure 2b, we depict conservative policy probabilities. Due to lower bound optimization, the actor prioritizes state-action subspaces that have already been explored and do not yield critic disagreement. Figure 2c illustrates optimistic policy probabilities. Despite having similar entropy levels, following the upper bound policy results in better coverage within critic disagreement regions.

distribution. Value is the expected discounted return from following the policy at a given state $V^\pi(s) = \int [R(s, a, s') + \gamma V(s')] \, ds'a$. Q-value is the expected discounted return from performing an action and following the policy thereafter $Q^\pi(s, a) = \int p(s') [R(s, a, s') + \gamma V(s')] \, ds'$. A policy is said to be optimal if it maximizes discounted return for starting state distribution. Actor-Critic (AC) for continuous action spaces performs simultaneous gradient-based learning of Q-values (*critic*) and policy (*actor*) that seeks local optimum of said Q-values (Silver et al., 2014; Ciosek & Whiteson, 2020). Critic parameters are updated by minimizing the SARSA temporal-difference variants (Sutton & Barto, 2018). Modern AC methods employ a variety of countermeasures to overestimation of Q-values, with bootstrapping using target network (Van Hasselt et al., 2016) and lower bound Q-value approximation (Fujimoto et al., 2018) being most prominent. Soft SARSA updates include policy stochasticity according to the following (Haarnoja et al., 2018):

$$\mathcal{L}^\theta = Q^\pi_\theta(s, a) - \left(R(s, a) + \gamma \left(Q^\pi_{lb}(s', a') - \alpha \log \pi_\phi(a'|s')\right)\right) \quad a' \sim \pi_\phi \quad s, a, s' \sim \mathcal{D} \quad (1)$$

Where $\pi_\phi$ is the actor; $Q^\pi_\theta$ is the critic; $Q^\pi_{lb}$ is the Q-value lower bound; $\alpha$ is the entropy temperature; and $\mathcal{D}$ denotes the experience buffer (Mnih et al., 2015). To achieve a locally optimal policy, the actor takes gradient steps aimed at maximizing the critic's lower bound (Moskovitz et al., 2021). The policy can use an exploration schedule (Fujimoto et al., 2018) or optimize its variance through soft policy improvement based on an entropy target (Haarnoja et al., 2018):

$$\mathcal{L}^\phi = -Q^\pi_{lb}(s, a) + \alpha \log \pi_\phi(a|s) \quad a \sim \pi_\phi \quad s \sim \mathcal{D} \quad (2)$$

As the actor models a parameterized distribution, gradients can be computed using the reparametrization trick (Kingma et al., 2014). When enforcing action domain constraints through hyperbolic tangent, minimizing policy log probabilities not only enhances exploration but also encourages the policy to maintain means within the non-saturated region of the hyperbolic tangent (Wang et al., 2020a). Additionally, the temperature can be automatically adjusted to ensure that average log probabilities match a specified target (Haarnoja et al., 2018):

$$\mathcal{L}^\alpha = -\alpha\left(\log \pi_\phi(a_i|s_i) + \mathcal{H}^*\right) \quad \alpha \in (0, \infty) \quad a \sim \pi_\phi \quad s \sim \mathcal{D} \quad (3)$$

Where $\mathcal{H}^*$ is the fixed entropy target which is often a function of action dimensionality. Contrary to fixed exploration scheduling, this method allows for heterogeneous variances across states. Given the optimization objective, this mechanism promotes exploration in states that offer lower Q-value

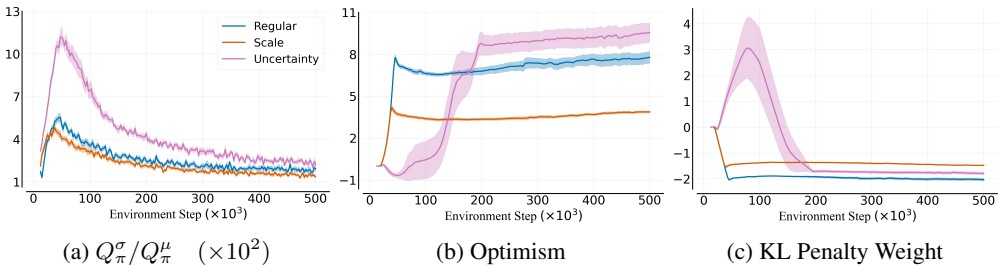

(a) $Q_\pi^\sigma / Q_\pi^\mu \quad (\times 10^2)$       (b) Optimism       (c) KL Penalty Weight

Figure 3: Varying reward scales, uncertainty levels, and Q-value non-stationarity pose challenges in setting fixed optimism ($\beta^{ub}$) and KL penalty weights. We examine three versions of the Cheetah Run task: regular (blue), equivalent to the vanilla DMC task; scale (orange), where we rescale Q-values by multiplying rewards; and uncertainty (pink), where we add Gaussian noise to rewards to increase aleatoric uncertainty. In Figure 3a), we observe how the ratios $Q^\sigma \pi / Q^\mu \pi$ change during training for these task variants. As training progresses, the divergence between policies maximizing Q-value lower and upper bounds decreases for any fixed $\beta^{ub}$. In Figure 3b), we illustrate the DAC optimism adjustment mechanism, which adapts $\beta^{ub}$ to achieve a desired empirical KL divergence between optimistic and conservative actors. This allows for task-dependent and phase-specific levels of optimism $\beta^{ub}$. Finally, Figure 3c presents DAC's KL penalty weight ($\tau$) on a logarithmic scale. Similarly to optimism, DAC adjusts the impact of the KL until the divergence reaches target levels.

gradients. For both actor and critic, an ensemble statistic of $k$ critic networks gives the Q-value lower bound. Most commonly, an ensemble of $k = 2$ is used. Then:

$$Q_{lb}^\pi(s,a) = \min\big(Q_\pi^1(s,a), Q_\pi^2(s,a)\big) = \underbrace{\frac{1}{2}\big(Q_\pi^1(s,a) + Q_\pi^2(s,a)\big)}_{\text{Mean}} - \underbrace{\frac{1}{2}\,|Q_\pi^1(s,a) - Q_\pi^2(s,a)|}_{\text{Standard Deviation}}$$

(4)

This observation generalized Q-value lower bound (Ciosek et al., 2019; Moskovitz et al., 2021):

$$Q_{lb}^\pi(s,a) = Q_\pi^\mu(s,a) + \beta^{lb}\, Q_\pi^\sigma(s,a) \tag{5}$$

Where $Q_\pi^\mu$ is the critic ensemble mean; $Q_\pi^\sigma$ is the critic ensemble standard deviation; and hyperparameter $\beta^{lb}$ controls the level of conservatism of the algorithm (ie. decreasing $\beta^{lb}$ leads to bigger penalization of critic disagreement). Setting $\beta^{lb} = -1$ is equivalent to the standard minimum of the two critics. Optimizing the actor with respect to the Q-value lower bound demotes state actions for which the critic ensemble disagrees. Such effect is referred to as pessimistic underexploration (showed in Figure 2) (Ciosek et al., 2019; Cetin & Celiktutan, 2023). OAC tackles the under-exploration by exploring according to an optimistic policy $\pi_\phi^o$, which is itself extracted from the conservative actor $\pi_\phi^c$. As such, OAC explores according to a transformed conservative policy $\pi_o$ given by the following Lagrangian:

$$\pi_e = \arg\max_{a \sim \pi_o} \mathbb{E}\, Q_{ub}^\pi(s,a) \quad \text{subject to} \quad D_{\text{KL}}\big(\pi_\phi^c(s) \,\|\, \pi_\eta^o(s)\big) \le \delta$$
$$with \quad Q_{ub}^\pi = a\nabla\, Q_\pi^\mu(s,a) + \beta^{ub}\, Q_\pi^\sigma(s,a)$$

(6)

Where $\delta$ is the boundary hyperparameter, $Q_{ub}^\pi$ is the Q-value upper bound approximated via a linear first-order Taylor series, and $\beta^{ub}$ hyperparameter controls the level of optimism. OAC exploration was shown to improve sample efficiency and performance as compared to SAC (Ciosek et al., 2019).

## 3    DECOUPLED ACTOR-CRITIC

Traditionally, AC algorithms use a single actor network for three main tasks: *exploration* (ie. sampling an action to add a transition to the experience buffer); *temporal-difference learning* (ie. sampling an action to calculate the TD target); and *evaluation* (ie. sampling an action to assess the

performance of an agent). Using a single actor for all tasks requires a delicate balance between optimism and conservatism. Exploration tends to favor optimistic behavior policies due to lower regret guarantees (Wang et al., 2020b), while TD learning leans towards conservatism due to the critic's tendency to overestimate (Hasselt, 2010). DAC addresses this dichotomy by introducing two distinct actor networks: an optimistic one and a conservative one. The optimistic actor is trained to maximize the upper bound of the Q-value and is exclusively used for exploration. On the other hand, the conservative actor is trained to maximize the lower bound of the Q-value and is employed for both TD learning and evaluation. By performing conservative Q-value updates on optimistic state-action samples, DAC achieves more effective exploration without the issue of Q-value overestimation.

---

**Algorithm 1** Decoupled Actor-Critic Step

---

1: **Input Models:** $\pi_\phi^c$ - conservative actor; $\pi_\eta^o$ - optimistic actor; $Q_\theta^\pi$ - critic ensemble; $Q_t^\pi$ - target critic; $\alpha$ - entropy temperature; $\beta^{ub}$ - optimism; $\tau$ - KL penalty weight;

2: **Input Hyperparameters:** $f_\sigma$ - variance multiplier described in Eq. 7; $x$ - copying frequency; $\mathcal{KL}^*$ - target KL divergence described in Eq. 9; $\beta_0^{ub}$ - initial $\beta^{ub}$; $\tau_0$ - initial $\tau$

3: $s', r, t = \text{ENV.STEP}(a) \quad with \quad a \sim f_\sigma(\pi_\eta^o(a|s))$ {sample from the optimistic actor}

4: $\text{BUFFER.ADD}(s, a, r, s', t)$

5: **if** $train\_step$ modulo $x = 0$; **then**

6: $\quad \eta \leftarrow \phi; \quad \beta^{ub}, \tau \leftarrow \beta_0^{ub}, \tau_0$ {copy conservative parameters; reinitialize $\beta^{ub}$ and $\tau$}

7: **end if**

8: **for** $i = 1$ **to** ReplayRatio **do**

9: $\quad s, a, r, s' \sim \text{BUFFER.SAMPLE}$

10: $\quad \theta \leftarrow \theta - \nabla_\theta \mathcal{L}^\theta(s, a, r, s', a') \quad with \quad a' \sim \pi_\phi^c$ {update critic according to Eq. 1}

11: $\quad \phi \leftarrow \phi - \nabla_\phi \mathcal{L}^\phi(s, a) \quad with \quad a \sim \pi_\phi^c$ {update conservative actor according to Eq. 2}

12: $\quad \eta \leftarrow \eta - \nabla_\eta \mathcal{L}^\eta(s, a) \quad with \quad a \sim f_\sigma(\pi_\eta^o)$ {update optimistic actor according to Eq. 7}

13: $\quad \alpha \leftarrow \alpha - \nabla_\alpha \mathcal{L}^\alpha$ {update entropy temperature according to Eq. 3}

14: $\quad \beta^{ub} \leftarrow \beta^{ub} - \nabla_\beta \mathcal{L}^\beta$ {update optimism according to Eq. 9}

15: $\quad \tau \leftarrow \tau - \nabla_\tau \mathcal{L}^\tau$ {update KL penalty weight according to Eq. 10}

16: $\quad Q_t^\pi = \text{POLYAK}(Q_\theta^\pi, Q_t^\pi)$ {standard Polyak averaging}

17: **end for**

---

The pseudo-code illustrates a single DAC training step, where changes with respect to SAC are colored. We summarize the most important novelties of the proposed algorithm: [1] *Decoupled Actors* - the conservative actor is used for TD learning (pseudo-code line 10) and the optimistic actor is used for exploration (pseudo-code line 3); [2] *Unique Variance* - the exploration policy can have a different level of entropy as compared to the TD learning policy (pseudo-code line 3); [3] *Optimistic Policy Objective* - the optimistic actor learns to maximize the regularized Q-value upper bound (pseudo-code line 12) with the levels of optimism and KL penalty weight adjusted such that the divergence target is met (pseudo-code lines 14 and 15). We describe all DAC modules in the following subsections and provide a detailed comparison to OAC in Appendix B.2.

## 3.1 CONSERVATIVE ACTOR, ENTROPY TEMPERATURE AND CRITIC

The conservative actor denoted as $\pi_\phi^c$, optimizes a standard soft policy target described in Equation 2. Using a soft policy target allows for state-dependent exploration and regularizes the policy such that the hyperbolic tangent output remains unsaturated. Furthermore, the non-zero variance of the conservative actor regularizes the critic TD learning. Since the data in $\mathcal{D}$ is collected exclusively by the optimistic actor, the conservative actor is updated fully off-policy. Following standard SAC, we update the entropy temperature and the critic via Equations 3 and 1 respectively. For both updates, the sampling is performed from the conservative actor $\pi_\phi^c$. Whereas in principle detaching exploration from exploitation allows for zero variance when sampling the TD targets, we find that including some levels of noise regularizes the critic. Finally, the critic uses layer normalization (Ba et al., 2016) before every activation, which we found to slightly increase the base agent's performance. We discuss the design choices in more detail in Appendix A.

## 3.2 OPTIMISTIC ACTOR

The optimistic actor, denoted as $\pi_\eta^o$, optimizes an optimistic policy objective defined as follows:

$$\mathcal{L}^\eta = -\underbrace{\left(Q_\pi^\mu(s,a) + \beta^{ub}\, Q_\pi^\sigma(s,a)\right)}_{\text{Q-value upper bound}} + \underbrace{\tau D_{\mathrm{KL}}\big(\pi_\phi^c(s) \parallel \pi_\eta^o(s)\big)}_{\text{Divergence penalty}} \quad a \sim f_\sigma(\pi_\eta^o) \quad s \sim \mathcal{D} \tag{7}$$

Where $\beta^{ub}$ is the optimism, $\tau$ is the KL penalty weight, $D_{\mathrm{KL}}$ is the KL divergence between the conservative and optimistic policies, and $f_\sigma$ is the standard deviation multiplier. Optimizing for the Q-value upper bound results in a policy that is optimistic in the face of uncertainty, but also promotes actions that generate critic disagreement. Since ensemble disagreement is often treated as a proxy for sample novelty (Yahaya et al., 2019; Han et al., 2021), following such a policy yields more diverse samples and as a result better coverage of the state-action space (Pathak et al., 2019; Lee et al., 2021). Whereas coverage is not explicitly optimized for in traditional RL, there is a growing body of research that hints toward the importance of data diversity in the context of RL (Xie et al., 2022; Foster et al., 2022; Zhan et al., 2022). KL divergence, the second objective term, regularizes the optimistic policy. While policies can be represented by various parameterized distributions, we implement both actors as simple diagonal normal distributions, transformed by the hyperbolic tangent activation. We compute the KL divergence in a closed form using the change of variables:

$$D_{\mathrm{KL}}\big(\pi_\phi^c(s) \parallel \pi_\eta^o(s)\big) = \sum_{i=1}^{|\mathcal{A}|} \left( \log \frac{\sigma_\phi^i(s)}{\sigma_\eta^i(s)} + \frac{\sigma_\eta^i(s)^2 + \big(\mu_\eta^i(s) - \mu_\phi^i(s)\big)^2}{2\,\sigma_\phi^i(s)^2} - \frac{1}{2} \right) \tag{8}$$

We derive the above statement in Appendix A. Using KL stabilizes the off-policy learning by ensuring that the sampled trajectories are probable under the conservative actor policy (Sutton & Barto, 2018; Ciosek et al., 2019). Secondly, it guarantees that the optimistic policy optimizes for a specified level of variance, which can be distinct from $\pi_\phi^c$. To this end, we define the function $f_\sigma$ as a simple variance multiplication. As such, the optimistic actor will have a standard deviation $f_\sigma$-times bigger than the conservative policy (this is implemented by simply multiplying the modeled standard deviations by $f_\sigma$). This mechanism allows for separate entropy for TD learning and exploration while retaining standard convergence guarantees of AC algorithms. In fact, as $\lim_{\mathcal{D}\to\infty} Q_\pi^\sigma(s,a) = 0$ (Van Hasselt et al., 2016), it follows that in the limit both actors recover a policy that differs only by $f_\sigma$. As shown in Figure 4, including the KL penalty is essential for the approach's success.

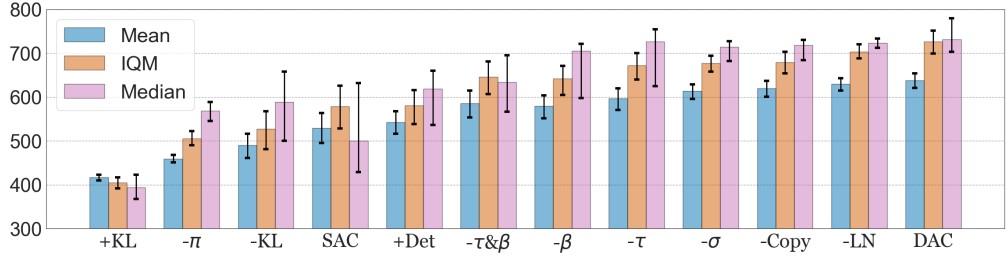

Figure 4: Evaluating the impact of various design choices in DAC. The ablated design choices include: $(+KL)$ KL penalty on both actors; $(-\pi)$ only optimistic actor; $(-KL)$ not using KL at all; $(+Det)$ a deterministic conservative actor; $(-\tau)$ a fixed value of $\tau$; $(-\beta)$ a fixed value of $\beta^{ub}$; $(-\tau\&\beta)$ fixed values of both; $(-\sigma)$ same variance on both actors; $(-copy)$ not copying parameters during the training; and $(-LN)$ DAC without layer normalisation. As follows, the application of KL is of great importance, with both using KL penalty on both agents and not using it at all leading to bad policies. $\beta^{ub}$ adjustment has more impact on the performance than $\tau$ adjustment. Finally, using DAC with parameter copying and layer normalization with DAC is beneficial. RR=3, 500k steps, 10 tasks, 10 seeds, and 95% bootstrapped CI. We detail the tested variations in Appendix G.

## 3.3 Adjustment of $\beta^{ub}$ and $\tau$

Since values of $Q_\pi^\mu$ and $Q_\pi^\sigma$ depend on reward scales, as well as aleatoric and epistemic uncertainty of the environment, the value of $\beta^{ub}$ cannot be easily set. Furthermore, as shown in Figure 3, fixed levels of $\beta^{ub}$ yield decreasing the impact of uncertainty on the optimistic policy. DAC leverages an observation that for $\beta^{ub} = -\beta^{lb}$ the optimistic actor recovers the objective of the conservative actor. Then, $\beta^{ub}$ can be defined such that the divergence between the conservative baseline policy and the optimistic policy reaches a desired level. To this end, implement a module that automatically adjusts the levels of optimism $\beta^{ub}$:

$$\mathcal{L}^{\beta^{ub}} = \left(\beta^{ub} + \beta^{lb}\right)\left(\frac{D_{\mathrm{KL}}\left(\pi_\phi^c(s) \parallel \pi_\eta^o(s)\right)}{|\mathcal{A}|} - \mathcal{KL}^*\right) \quad \beta^{ub} \in (\beta^{lb}, \infty) \quad s \sim \mathcal{D} \quad (9)$$

Where $\mathcal{KL}^*$ is the KL divergence target between the optimistic and transformed conservative policies, $D_{\mathrm{KL}}$ is the empirical KL divergence, and $|\mathcal{A}|$ is the action dimensionality. If the empirical KL divergence is bigger than the KL target, then $\beta^{ub}$ is reduced with a limit at $\beta^{lb}$. On the other hand, if the empirical KL divergence is smaller than the target, then $\beta^{ub}$ is increased with a limit at $\infty$. This update mechanism allows us to define optimism level as a divergence between optimistic and conservative policies. We update the KL penalty weight $\tau$ in the opposite direction:

$$\mathcal{L}^\tau = -\tau\left(\frac{D_{\mathrm{KL}}\left(\pi_\phi^c(s) \parallel \pi_\eta^o(s)\right)}{|\mathcal{A}|} - \mathcal{KL}^*\right) \quad \tau \in (0, \infty) \quad s \sim \mathcal{D} \quad (10)$$

By dividing by $|\mathcal{A}|$ we allow for divergence per degree of freedom. As $\tau$ is increased when the empirical KL target is bigger than the desired KL target, DAC can regularize the divergence between two actors even if the $\beta^{ub}$ is at its negative limit. Conversely, an automatic reduction of $\tau$ accompanies the increase of $\beta^{ub}$ if reaching the divergence limit proves challenging. This adaptive approach, as illustrated in Figure 3c, accommodates different scales of Q-values and contrasts with setups like OAC, where optimism is predefined by fixing $\beta^{ub}$ at a specific value. However, if the adjustment mechanism operates too slowly, the KL penalty may not be effectively enforced during training, potentially causing the two agents to diverge. This divergence can result in fully off-policy learning, insufficient coverage in the conservative policy region, and ultimately, suboptimal agent performance. We believe that this issue may be connected to the deadly triad (Thrun & Schwartz, 2014; Sutton & Barto, 2018) and recent findings highlighting the limitations of fully off-policy learning, such as in the tandem setting (D'Oro et al., 2022). To mitigate the divergence problem, we observed that initializing both agents with identical parameter values ($\phi_0 = \eta_0$) makes them less likely to diverge. Additionally, during training, we employ a hard parameter copy of the conservative actor and reinitialize the optimistic actor with copies of these parameters.

## 4 Experiments

We build our experiments on JaxRL code base (Kostrikov, 2021). Since all considered algorithms other than SAC are extensions of thereof, we fix the pool of common hyperparameters on values that are known to work well with SAC (Nikishin et al., 2022; D'Oro et al., 2022). All algorithm-specific hyperparameters have fixed values between low and high replay ratio settings and are reported in Appendix E. Similarly, all algorithms except RedQ use the same network architectures and a standard

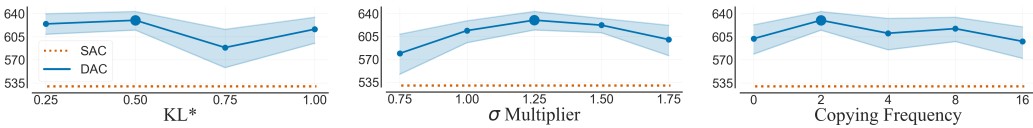

Figure 5: Impact of DAC hyperparameters on the final performance. All tested setups outperformed baseline SAC (orange), demonstrating DAC robustness and stability. The thick dot is the configuration used in the main experiment. $500k$ steps, 10 tasks, 10 seeds, mean and 95% bootstrapped CI.

ensemble of two critics (Fujimoto et al., 2018; Haarnoja et al., 2018; Ciosek et al., 2019; Moskovitz et al., 2021; Cetin & Celiktutan, 2023). For all experiments, we report robust evaluation statistics generated via the RLiable package (Agarwal et al., 2021). The results for the main experiments are presented in Figures 1, 6 & 7. Additionally, we run ablations on various design choices and hyper-parameters which we report in Figures 4 & 5. We provide further experimental results in Appendix D and information on the experimental settings in Appendix G

**Low and High Replay** We consider a set of 10 proprioceptive DeepMind Control Suite (DMC) tasks (Tassa et al., 2018) listed in Appendix F for which we run experiments in low and high replay regimes. In both, we use $10^6$ environment steps for each task and algorithm. Firstly, we consider a low replay ratio of 3 gradient steps per environment step. Such a low replay does not induce loss of plasticity or overfitting in tested algorithms (Nikishin et al., 2022; Li et al., 2022). As such, no parameter resets are required (D'Oro et al., 2022). We consider the following baselines: OAC (Ciosek et al., 2019); ND-TOP (Moskovitz et al., 2021); SAC (Haarnoja et al., 2018); and TD3 (Fujimoto et al., 2018). Furthermore, we consider a high replay of 15 gradient steps per environment step. Such replay is known to degenerate the performance of most algorithms unless regularization is used (Chen et al., 2020; D'Oro et al., 2022). To this end, all algorithms perform full-parameter resets in $50000th$ step, as well as every 250000 environment steps (Nikishin et al., 2022; D'Oro et al., 2022). In this setup, we consider SR-SAC (D'Oro et al., 2022), SR-TOP and SR-OAC.

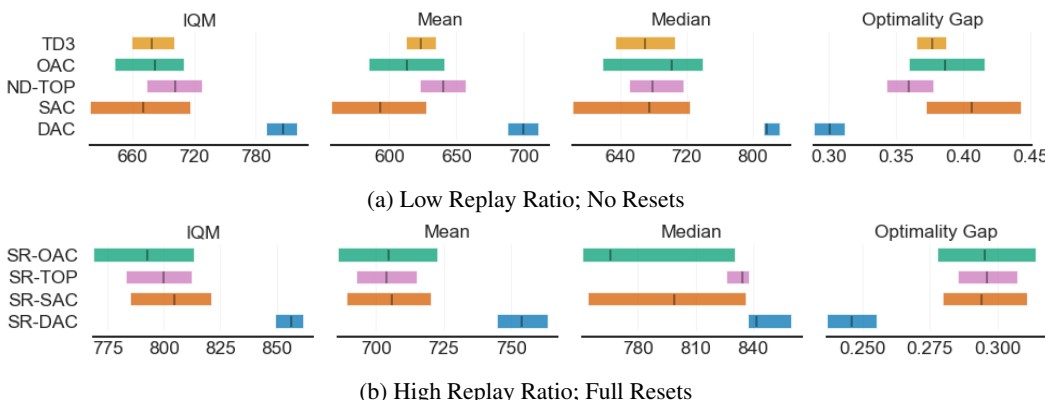

(a) Low Replay Ratio; No Resets

(b) High Replay Ratio; Full Resets

Figure 6: RLiable results for two regimes. Tasks used in the evaluation are listed in Appendix F. DAC achieves the best final performance, with a pretty sizeable performance gap in the low replay ratio regime. We use 10 tasks, 10 random seeds, and $10^6$ environment steps. The bars indicate the 95% bootstrapped CI. We provide detailed training curves in Appendix H.

We find that low replay DAC achieves significantly better performance than the baseline algorithms (Figures 1a and 6a). Notably, low replay DAC matches the performance of SR-SAC (Scaled-by-Resetting SAC), despite SR-SAC utilizing 5-times bigger replay and resets (Figure 1c). Similarly, scaled-by-resetting DAC (SR-DAC) outperforms the baseline algorithms in the high replay regime. As shown in Figures 1b and 6b, SR-DAC achieves better performance than the baseline algorithms and significantly surpasses the state-of-the-art model-free SR-SAC.

**Dog Domain** We assess DAC performance on three tasks from the complex dog domain. The domain features a significantly larger action space than humanoid (31 dimensions compared to humanoid's 21). and remains unsolved by a model-free agent with proprioceptive input. Recently, it was shown that a model-based agent TD-MPC achieves substantial improvements to SAC on the dog tasks (Hansen et al., 2022). As such, we evaluate whether DAC can find policies better than SAC and if it is capable of surpassing the performance of model-based TD-MPC on dog tasks. As TD-MPC uses a variety of annealing mechanisms, a prioritized experience buffer, and a much greater number of parameters, we decided to add one hidden layer to each network in DAC and increase the copying frequency of the optimistic agent. We refer to such changed implementation as SR-DAC+. We run the algorithm for $30^6$ environment steps and compare its performance against both SR-SAC and TD-MPC. We detail this experimental setting in Appendix G and E. We summarize the results in Figure 7 and provide an additional comparison against TD-MPC in Appendix D.

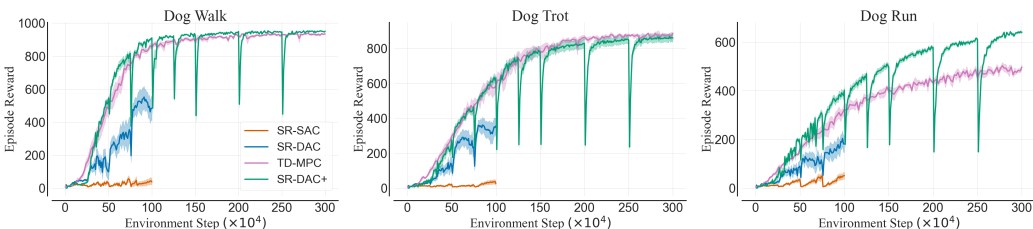

Figure 7: We find that SR-DAC achieves substantially better performance than SR-SAC, and SR-DAC+ outperforms TD-MPC. To the best of our knowledge, SR-DAC+ achieves the highest recorded performance on the dog tasks of a model-free RL algorithm. SR-DAC and SR-SAC use 10 seeds each, and SR-DAC+ and TD-MPC use 5 seeds each. Mean and 95% bootstrapped CI.

Whereas SR-SAC is unable to find a performing policy in the dog domain (a result consistent with the evaluations performed in Hansen et al. (2022)), DAC is able to party solve all of the dog tasks. Furthermore, we find that the SR-DAC+ matches the state-of-the-art model-based TD-MPC.

## 5 LIMITATIONS

Besides the limitations that DAC inherits from the soft actor-critic algorithmic family, DAC divergence minimization presents unique optimization challenges. Unlike typical uses of KL divergence, where the target distribution remains fixed (eg. Variational Autoencoders (VAE) (Kingma et al., 2014)), DAC deals with a constantly evolving policy that is continually improving. Consequently, the optimistic actor needs to keep up with the conservative actor's changes. As depicted in Figure 4, DAC heavily relies on maintaining a low divergence between the actors. While DAC adjustment mechanisms proved effective in the tested environments, there is no guarantee that they will suffice in more complex ones. The second drawback of DAC lies in its inherent use of two actor networks, which results in slightly increased memory and computational demands compared to the standard Soft Actor-Critic (SAC) approach. In practice, the wall-clock time of DAC is around 10% greater than that of SAC and is indistinguishable from the overhead induced by OAC, which requires additional backpropagation through the critic ensemble. Moreover, since DAC initializes both actors with identical parameters, they must share the same network architecture. However, as indicated by Figure 4, simply copying parameters between them offers only minimal performance enhancement. In light of this, we believe that the necessity for identical architectures can be mitigated by employing techniques like delayed policy updates (Fujimoto et al., 2018) or by learning rate scheduling.

## 6 CONCLUSIONS

In this paper, we introduced DAC, an off-policy algorithm that leverages two distinct actors trained via specialized objectives. One actor, known as the conservative actor, is dedicated to TD learning and evaluation tasks, while the other, the optimistic actor, is used in exploration. This allows DAC to perform conservative Q-value updates at optimistic state-action samples. As a result, DAC directly addresses the optimism-pessimism dilemma commonly encountered in Actor-Critic agents. To evaluate the effectiveness of the proposed method, we conducted experiments on a set of 10 complex locomotion tasks, considering two different replay ratio regimes. Our results demonstrated that DAC significantly outperforms established benchmark algorithms in terms of both performance and sample efficiency. To assess the impact of individual DAC components, we conducted extensive ablation studies consisting of over 2000 runs. Finally, we showcased the robustness of DAC across a range of hyperparameter settings, underscoring its suitability for practical applications.

### REPRODUCIBILITY

We provide the DAC implementation, results and scripts used to generate the results at the following URL. For pseudo-code, implementation details, or additional information about specific design choices, please refer Appendix A. We describe experimental settings in Section 4 and Appendix G. Finally, we point the reader towards Appendix E for the hyperparameters used in experiments

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

## APPENDIX CONTENTS

We divide the Appendix into the following sections:

1. Further Description of DAC (Appendix A) — We provide additional rationale for certain design choices in DAC, as well as additional implementation details.
2. Related Work (Appendix B) — We expand on the works related to DAC and provide a detailed comparison to the mechanisms leveraged by Optimistic Actor-Critic (OAC).
3. Future Work (Appendix C) – we discuss a variety of research directions related to DAC
4. Additional Experiments (Appendix D) — We report the additional tests of decoupled architecture in the context of value overestimation; performance impact of action repeat; and additional evaluation against TD-MPC.
5. Hyperparameter Settings (Appendix E) — We list all hyperparameter settings required for the reproduction of the experiments.
6. Tested Environments (Appendix F) — We list all DMC tasks using low replay, high replay, and DreamerV3 experiments.
7. Experimental Settings (Appendix G) — We provide a detailed description of all experimental settings used in the paper.
8. Learning Curves (Appendix H) — We share learning curves for the low replay, high replay, and DreamerV3 comparisons.

## A    FURTHER DESCRIPTION OF DAC

### A.1    SOFT CONSERVATIVE ACTOR

When using a Tanh-Normal distribution to represent a policy, the soft policy learning (Haarnoja et al., 2018) serves two key purposes: *enforcing entropy*; and *enforcing non-saturation*.

**Enforcing entropy**    soft policy learning ensures a specific level of policy entropy, thereby ensuring exploration during the learning (Haarnoja et al., 2018). Additionally, optimizing the objective function that balances entropy maximization and Q-values, enables policies to exhibit state-dependent entropy promoting more exploration in low Q-value states. This stands in contrast to traditional exploration methods employed by algorithms such as TD3, or TOP, where the policy maintains a constant entropy across different states.

**Enforcing non-saturation**    since the Tanh function saturates, increasing the distance between the unsquashed distribution expected value and zero naturally decreases the distribution variance after squashing, as illustrated in Figures 8a and 8b. Consequently, when employing soft learning, the agent experiences a loss when shifting the policy away from zero. As depicted in Figures 8c and 8d, this results in policies that concentrate within the non-saturated region of Tanh (Wang et al., 2020a) and thus avoid bang-bang behaviour (Seyde et al., 2021).

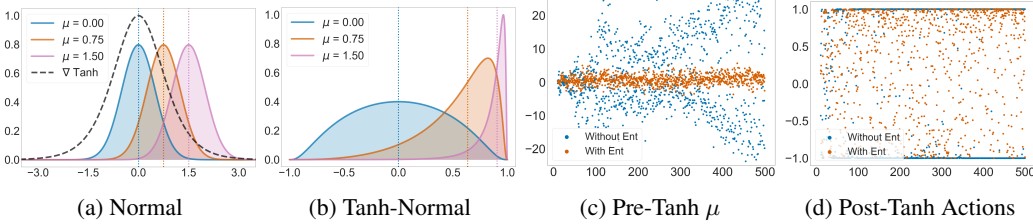

(a) Normal          (b) Tanh-Normal          (c) Pre-Tanh $\mu$          (d) Post-Tanh Actions

Figure 8: Soft policy learning enforces prevents policy saturation. Figures 8a and 8b show three Gaussians and the Tanh transformed counterparts. Before Tanh transformation, all policies have equal variance (8a). However, since Tanh is nonlinear, applying the transformation changes the policy variance depending on the location of the mean (8b). Because of this effect, when using soft policy learning, the agent incurs loss when moving along the saturated portions of the Tanh activation function. Figures 8c and 8d show policy means and executed actions for SAC with and without soft policy learning on a Humanoid-Stand task. As follows, SAC without the entropy objectives follows a bang-bang policy, whereas regular SAC anchors the policy within the non-saturated portion of Tanh. To this end, soft policy learning reduces the risk of a bang-bang policy.

When designing DAC, we were interested in enforcing entropy and non-saturation on both conservative and optimistic actors. DAC achieves this by using soft policy learning on the conservative actor (enforcing entropy and non-saturation) and using KL divergence on the optimistic actor (enforcing similarity between the optimistic and conservative actors).

### A.2    OPTIMISTIC POLICY LEARNING AND PARAMETER COPYING

We define the optimistic policy learning objective as follows:

$$\mathcal{L}^{\eta} = -\underbrace{\left(Q_{\pi}^{\mu}(s,a) + \beta^{ub}\, Q_{\pi}^{\sigma}(s,a)\right)}_{\text{Q-value upper bound}} + \underbrace{\tau D_{\text{KL}}\big(\pi_{\phi}^{c}(s) \,\|\, \pi_{\eta}^{o}(s)\big)}_{\text{Divergence penalty}} \quad a \sim f_{\sigma}(\pi_{\eta}^{o}) \quad s \sim \mathcal{D} \tag{11}$$

For an explanation of each symbol, please refer to Section 3.2. In our approach, we emphasize explicit upper bound optimization as the core principle of the optimistic policy objective. This aspect of the objective aims to find a policy with minimal regret. It's important to note that the optimistic actor is exclusively employed for exploration purposes, ensuring that it doesn't interfere

with Q-value learning. This approach stands in contrast to the pessimistic lower bound approach used in baseline algorithms such as SAC or TOP.

Furthermore, the optimistic policy objective features a KL divergence penalty. As shown in ablation studies in Figure 4, such a penalty is of crucial importance for DAC performance. The penalty is applied only to the optimistic actor and serves multiple purposes:

1. Reducing the degree of off-policy learning - since the exploration is done solely by the optimistic actor, the conservative actor is updated fully off-policy (ie. on transitions sampled from the optimistic policy). This practice can lead to unstable learning, a problem known as the "deadly triad" (Sutton & Barto, 2018). By incorporating KL divergence into the optimistic objective, we ensure that the transitions sampled from the optimistic policy align with the probabilities expected under the conservative policy. This serves to mitigate the extent of off-policy learning.

2. Anchoring the adjustment mechanisms - as discussed in Section 3.3, the automatic adjustment of $\beta^{ub}$ (optimism) and $\tau$ (KL penalty weight) is anchored on some desired divergence value (ie. it is designed to adjust $\beta^{ub}$ and $\tau$ such that the divergence target is met). To this end, without including the divergence penalty we would not be able to use the adjustment mechanisms as designed.

3. Enforcing non-saturation of the optimistic policy - by minimizing KL between the conservative actor and itself, the optimistic actor is enforced to mimic a policy that is trained via soft policy learning. As such, the optimistic actor becomes disincentivized from saturating.

As discussed in Section **??**, any differentiable divergence or distance function could be used in place of the KL. What makes KL divergence appealing is that it has a closed-form solution for any invertible and differentiable transformation of given distributions. Although it is a well-known result, we leave it below for completeness. We denote $x$ as samples from policy distributions before applying the Tanh activation, $p_c$ and $p_o$ as conservative and optimistic distributions on $x$, Tanh application as $y = h(x)$ and the resulting distributions as $\pi_c, \pi_o$. Then leveraging the change of variables formula:

$$
\begin{aligned}
D_{\mathrm{KL}}\big(p_c \| p_o\big) &= \int_{-\infty}^{\infty} p_c(x) \log \frac{p_c(x)}{p_o(x)} dx \\
&= \int_{-\infty}^{\infty} \pi_c(h(x)) \,|\tfrac{dy}{dx}(x)| \, \log \frac{\pi_c(h(x)) \,|\tfrac{dy}{dx}(x)|}{\pi_o(h(x)) \,|\tfrac{dy}{dx}(x)|} dx \\
&= \int_{-\infty}^{\infty} \pi_c(y) \log \frac{\pi_c(y)}{\pi_o(y} dy = D_{\mathrm{KL}}\big(p_c \| p_o\big)
\end{aligned}
\tag{12}
$$

The above holds true for any distributions $p_c$ and $p_o$. Since we assume them to be diagonal Gaussian, it follows that:

$$
D_{\mathrm{KL}}\big(p_c \| p_o\big) = D_{\mathrm{KL}}\big(\pi_c \| \pi_o\big) = \left( \log \frac{\sigma_c}{\sigma_o} + \frac{\sigma_o^2 + (\mu_o - \mu_c)^2}{2\sigma_c^2} - \frac{1}{2} \right)
\tag{13}
$$

Which concludes the derivation.

### A.3   CRITIC REGULARIZATION

Recently, there has been a surge of works exploring the importance of critic regularization (Liu et al., 2020; Laskin et al., 2020; Hiraoka et al., 2021; Gogianu et al., 2021; Li et al., 2022; D'Oro et al., 2022). Whereas there is still a lot to understand about the interplay of TD learning and network regularization, it is clear that a regularized critic allows a higher replay ratio to be used (Hiraoka et al., 2021; Li et al., 2022; D'Oro et al., 2022). In this paper, we explore only two regularization methods: layer normalization (Ba et al., 2016) and full-parameter resets (Nikishin et al., 2022). Given the effectiveness of the only regularization we tested, we hypothesize that experimenting with

methods like weight decay (Krogh & Hertz, 1991), spectral normalization (Gogianu et al., 2021) or dropout (Srivastava et al., 2014) could further improve DAC performance. As follows from the Figure 9, layer normalization has positive effects on DAC performance, particularly in the high replay ratio regime. This suggests, somewhat surprisingly, that layer normalization might have some synergy effect with full-parameter resets.

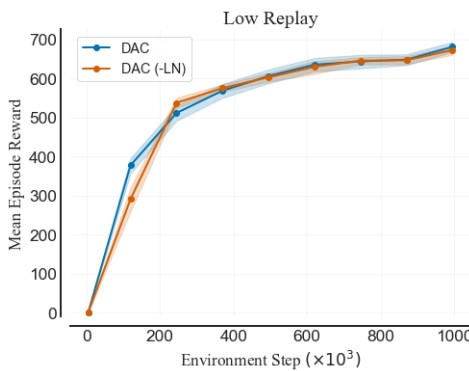 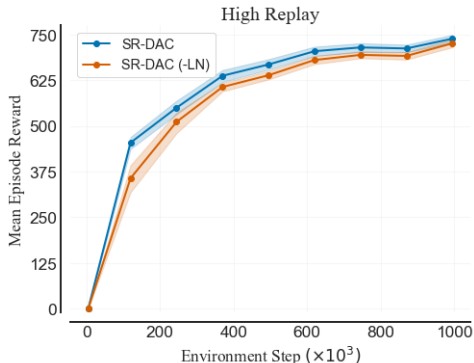

Figure 9: Layer normalization applied to critic slightly improves the performance of DAC. 10 tasks, 10 seeds, mean and 95% bootstrapped CI.

## B  RELATED WORK

### B.1  UNCERTAINTY EXPLORATION AND KL CONSTRAINED POLICIES

The exploration-exploitation dilemma has been the subject of extensive research. One prominent principle that has emerged in addressing this dilemma is Optimism in the Face of Uncertainty (OFU) (Auer et al., 2002; Filippi et al., 2010; Ciosek et al., 2019), which prioritizes actions with a balance of high expected rewards and uncertainty. Whereas OFU has been extensively studied in the tabular and bandit RL setting (Auer et al., 2002; Garivier & Moulines, 2011; Kaufmann et al., 2012), it has not yet become as standard in deep RL. However, it has been shown that DQN ensembles used for uncertainty-driven updates can provide performance improvements (Osband et al., 2016; Chen et al., 2017; Osband et al., 2018; Lee et al., 2021). Similarly, OAC (Ciosek et al., 2019) and TOP (Moskovitz et al., 2021) leverage uncertainty estimates over the state-action value function for exploration, albeit still using one conservative actor.

The optimism-driven exploration was also considered for model-based agents. Sekar et al. (2020) and Seyde et al. (2022) consider exploration driven by reward model ensemble. Similarly to DAC, Seyde et al. (2022) considers using an optimistic upper bound exploration policy and a distinct exploitation policy. Furthermore, agents like RP1 (Ball et al., 2020) leverage reward uncertainty despite access to the nominal rewards. Finally, a variety of agents that leverage MCTS have been proposed (Silver et al., 2017; Schrittwieser et al., 2020; Zawalski et al., 2022). On a different note, DAC might seem similar to natural actor-critic (Amari, 1998; Kakade, 2001; Peters & Schaal, 2008; Schulman et al., 2015) due to the KL constraint in actor optimization. Whereas the natural policy gradient uses KL to enforce the similarity between policies resulting in subsequent updates, DAC leverages KL to constrain the optimistic policy to be within some range from the conservative policy that is used for TD learning. This allows for optimistic exploration without Q-value overestimation.

### B.2  COMPARISON TO OAC

DAC leverages two policies: a conservative one used for sampling the temporal-difference target and evaluation; and an optimistic one used for sampling transitions added to the experience buffer. Similarly to DAC, OAC performs evaluation and Bellman backups according to a conservative lower bound policy. However, DAC differs from OAC on three main design choices: *how to model the optimistic policy*; *how to constraint the optimistic policy*; and *how to set the level of optimism* $\beta^{ub}$.

**How to model the optimistic policy**   OAC models the optimistic policy by combining conservative policy with the linear approximation of Q-value upper bound, and as such uses one actor network. The linear approximation combined with constrained optimization results in simplistic solutions along the constraint boundary (Protter et al., 2012). As such, OAC's applicability is limited to small $\delta$ values due to the Taylor theorem. In contrast to that, DAC uses two actors. Modeling the second policy via an actor network allows for exploration policies that are far more complex than a linear approximator. Whereas this introduces a computational cost, employing techniques like delayed policy updates can result in costs smaller than that of OAC.

**How to constraint the optimistic policy**   OAC enforces a hard KL constraint by directly solving a Lagrangian. Since the Q-value upper bound is approximated via a linear function, the solution is placed on the constraint boundary unless the slope is zero (Protter et al., 2012). In contrast, DAC imposes KL as a soft constraint. Paired with the neural network approximator, this allows DAC to balance the KL with potential gains to the upper bound and generate complex exploration policies.

**How to set the level of optimism $\beta^{ub}$**   Finally, OAC treats $\beta^{ub}$ as a hyperparameter which is fixed during the training. Since values of $Q_\pi^\mu$ and $Q_\pi^\sigma$ depend on reward scales, as well as aleatoric and epistemic uncertainty of the environment, the value of $\beta^{ub}$ has to be searched per task. Furthermore, as shown in Figure 3, fixed levels of $\beta^{ub}$ yield decreasing the impact of uncertainty on the optimistic policy. DAC leverages that the desired level of optimism can be defined through divergence between the conservative baseline policy and the optimistic policy optimizing objective related to $\beta^{ub}$. Such definition allows for dynamics adjustment of both $\beta^{ub}$ and the KL penalty weight $\tau$.

### B.3   REPLAY RATIO SCALING

Increasing the replay ratio was the key ingredient in increasing the sample efficiency and final performance of many recent RL agents (Janner et al., 2019; Chen et al., 2020; Hiraoka et al., 2021; Nikishin et al., 2022; Schwarzer et al., 2023). Moreover, many of the novelties brought by those agents were focused on stabilizing the learning given the high-replay regime (Janner et al., 2019; Chen et al., 2020; Hiraoka et al., 2021). A particularly interesting technique is scaling-by-resetting, as it is applicable to most off-policy RL algorithms. DAC improvements are largely orthogonal to the direction of increasing the replay ratio, but DAC also takes advantage of the scaling-by-resetting technique. However, as the available replay ratio is often an exogenous constraint (eg. when running models on the robot), we believe that testing new algorithms in both high and low replay is valuable.

## C   FUTURE WORK

One of the critical functions of DAC is limiting the divergence between the two actors (see Figure 4). This aspect raises an interesting question about the potential tradeoff between the performance gains achieved by adhering to a low-regret optimistic policy and the performance losses incurred from fully off-policy updates. To control the divergence between the two actors, we employ a KL penalty, although we believe that alternative divergence or distance metrics could also be effective. The main reason for using KL divergence in our implementation of DAC is that it is known to have a closed-form solution for Tanh-Normal distributions which we use to model both policies. We think that implementing DAC with regularization other than KL might result in better learning stability.

Novel mechanisms used by DAC are orthogonal to many recent improvements in DRL. As such, investigating synergies between DAC and techniques like receding TD horizon (Kearns & Singh, 2000; Schwarzer et al., 2023), critic regularization (Gogianu et al., 2021), discount factor annealing (Ye et al., 2021; Schwarzer et al., 2023), AVTD (Li et al., 2022), TOP (Moskovitz et al., 2021) or increasing model size (Schwarzer et al., 2023; Hafner et al., 2023) might improve DAC performance. Furthermore, distributional critics offer a capability to directly model both aleatoric and epistemic uncertainties (Bellemare et al., 2017; Dabney et al., 2018; Moskovitz et al., 2021). We think that this aligns with the DAC, as it builds policies leveraging epistemic uncertainty. Similarly, expanding the size of the critic ensemble could lead to synergies and improvements surpassing those achieved by conventional ensemble AC approaches (Lee et al., 2021; Januszewski et al., 2021). Finally, as shown in Figure 4, the deterministic version of DAC underperforms its stochastic counterpart. Investigating the factors contributing to this difference in performance is a compelling avenue for research.

## D    ADDITIONAL EXPERIMENTS

### D.1    CRITIC OVERESTIMATION

In this subsection, we investigate whether the decoupled architecture (using a conservative policy for TD updates and optimistic policy for exploration) indeed mitigates Q-value overestimation characteristic for non-conservative updates (Fujimoto et al., 2018; Moskovitz et al., 2021). As such, we compare the environment returns to the returns implied by the critic of three actor-critic architectures.

1. Conservative - SAC agent performing conservative ($\beta^{lb} = -1$)
2. Optimistic - SAC agent performing optimistic updates ($\beta^{lb} = 1$)
3. Decoupled - DAC performing conservative updates on the critic and the conservative actor; optimistic updates on the optimistic actor ($\beta^{lb} = -1$)

We conduct such measurements on 3 DMC environments of varying difficulty: hopper-hop; quadruped-run; and humanoid-walk. The results are summarized in Figure 10. Assuming deterministic transitions, the soft Q-value modeled by the critic is equal to the sum of discounted returns and policy entropy. As such, to calculate the bias we have to subtract the entropy term from the critic's output. We estimate the entropy sum term by a geometric sum of negative average policy log probabilities. Then, we again use the geometric sum and average recorded return to estimate the real Q-values.

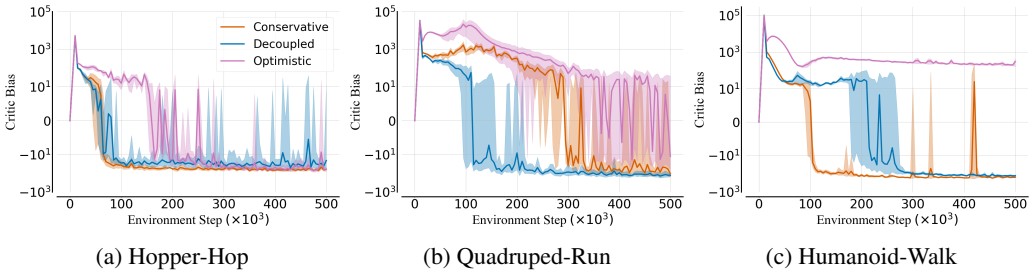

|     |     |     |
| --- | --- | --- |
| (a) Hopper-Hop | (b) Quadruped-Run | (c) Humanoid-Walk |

Figure 10: We evaluate critic overestimation associated with conservative (SAC with $\beta^{lb} = -1$), optimistic (SAC with $\beta^{lb} = 1$) and decoupled agents (DAC). We observe that the baseline conservative SAC tends to underestimate the returns. 5 seeds, mean, and 3 std.

We find that the decoupled architecture indeed prevents the overestimation associated with optimistic TD updates. Interestingly, we observe that both conservative and decoupled architectures tend to underestimate the returns. This observation is consistent with the earlier results pointing towards over-conservatism of clipped double Q-learning (Moskovitz et al., 2021; Cetin & Celiktutan, 2023).

### D.2    ACTION REPEAT EXPERIMENT

Furthermore, we validate the findings of our main experiment under a different setting of environment Action Repeat (AR). AR is a key environment parameter that reduces the agent's action frequency while preserving the underlying environment dynamics. It achieves this by having the agent repeat the same action for a set number of consecutive time steps instead of selecting a distinct action at each time step. When AR is employed, the transitions resulting from these repeated actions are merged into a single transition, with rewards summed and the final next state used. Consequently, higher AR values decrease the length of trajectories, thereby enhancing the diversity of data the agent encounters within a fixed number of interactions. Action repeat is known to impact the sample efficiency and the final performance of agents, and in the case of DMC, a variety of AR settings have been considered. Common choices include repeats of: 1 (ie. no repeat) (D'Oro et al., 2022; Li et al., 2022); 2 (ie. single repeat) (Hafner et al., 2019a; Nikishin et al., 2022); or varying setting between the tasks (Hafner et al., 2019b; Yarats et al., 2020). As noted in Appendix E, we use an action repeat of 2 throughout our experiments. However, for a matter of completeness, we include an evaluation for AR of 1 on our benchmark of 10 DMC tasks in Figure 11 below.

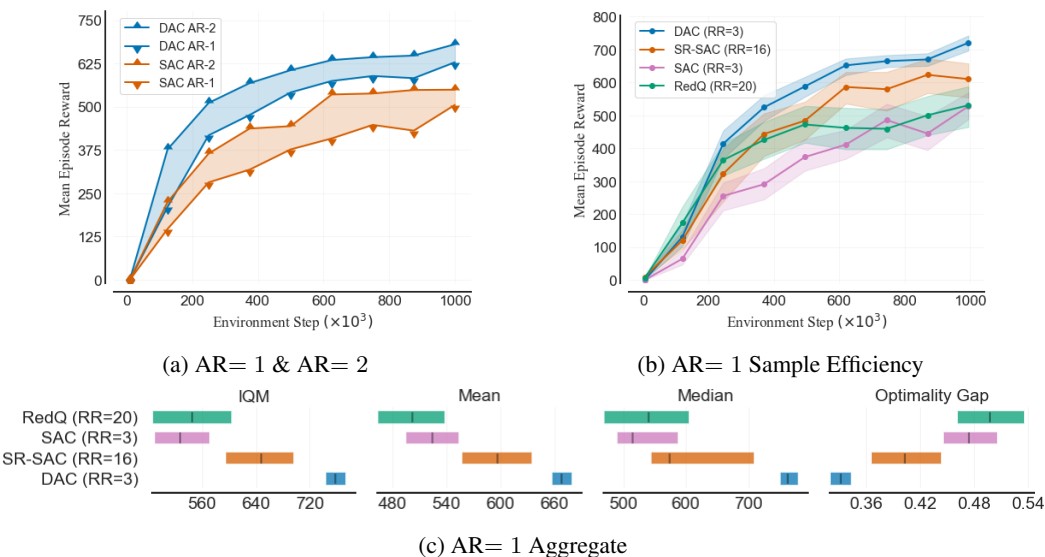

(a) AR= 1 & AR= 2       (b) AR= 1 Sample Efficiency

(c) AR= 1 Aggregate

Figure 11: DAC consistently improves performance across various AR settings. In Figure 11a, we compare SAC and DAC under low replay conditions for two AR configurations. As evident, higher AR values generally lead to superior performance for both algorithms, with DAC significantly outperforming SAC in both scenarios. Figures 11b and 11c present additional results when AR is set to 1. RedQ performs similarly to a well-tuned SAC with a replay of 3, whereas DAC outperforms both. We include SR-SAC and RedQ as reported in D'Oro et al. (2022). 10 seeds (except for SR-SAC which has 5 seeds), 10 tasks, and 95% bootstrapped CI.

## D.3  TD-MPC EVALUATION

Figure 12 summarizes additional results for the TD-MPC comparison. We list all hyperparameter differences to regular DAC implementation in Table 2..

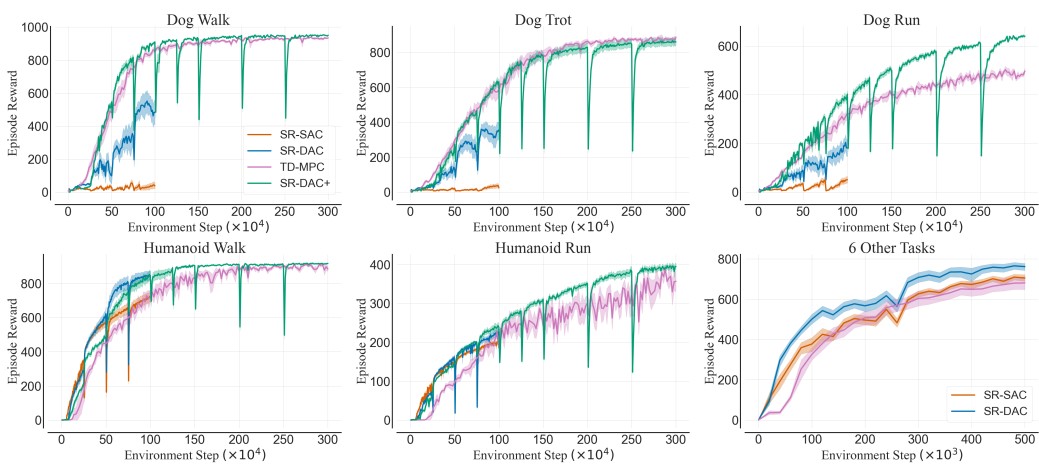

Figure 12: We compare SR-DAC against SR-SAC and TD-MPC. We find that SR-DAC achieves substantially better performance than SR-SAC, whereas SR-DAC+ performs better than TD-MPC. The lower right graph shows the average from the following 6 tasks: hopper-hop; cheetah-run; quadruped-run; humanoid-stand; fish-swim and cartpole-swingup_sparse. SR-DAC and SR-SAC use 10 seeds for each task, and SR-DAC+ and TD-MPC use 5 seeds for each task.

# E HYPERPARAMETER SETTINGS

In all experiments, all algorithms use the same network architectures and a standard ensemble of two critics (except RedQ, which by design is required to increase the ensemble size). We fix the shared pool of hyperparameters to single values, which we list under the category share in the table below. We use the hyperparameter values which are known to work well with SAC (Nikishin et al., 2022; D'Oro et al., 2022).

Table 1: Hyperparameter values used in the experiments.

| HYPERPARAMETER | NOTATION | VALUE |
|---|---|---|
| NETWORK SIZE | NA | $(256, 256)$ |
| ACTION REPEAT | NA | 2 |
| OPTIMIZER | NA | ADAM |
| LEARNING RATE | NA | $3e - 4$ |
| BATCH SIZE | $B$ | $3e - 4$ |
| DISCOUNT | $\gamma$ | 0.99 |
| INITIAL TEMPERATURE | $\alpha_0$ | 1.0 |
| INITIAL STEPS | NA | 10000 |
| TARGET ENTROPY | $\mathcal{H}^*$ | $|\mathcal{A}|/2$ |
| POLYAK WEIGHT | $\tau$ | 0.005 |
| PESSIMISM | $\beta^{lb}$ | $-1.0$ |
| TD3 | | |
| POLICY UPDATE DELAY | NA | 2 |
| EXPLORATION $\sigma$ | NA | 0.5 |
| TARGET POLICY $\sigma$ | NA | 0.2 |
| EXPLORATION NOISE CLIP | NA | $[-0.5, 0.5]$ |
| OAC / SR-OAC | | |
| OPTIMISM | $\beta^{ub}$ | 4.36 |
| KL DIVERGENCE CONSTRAINT | $\delta$ | 3.69 |
| ND-TOP / SR-TOP | | |
| OPTIMISM ARMS | NA | $[0, 1]$ |
| BANDIT LEARNING RATE | NA | 0.1 |
| POLICY UPDATE DELAY | NA | 2 |
| EXPLORATION $\sigma$ | NA | 0.5 |
| TARGET POLICY $\sigma$ | NA | 0.2 |
| EXPLORATION NOISE CLIP | NA | $[-0.5, 0.5]$ |
| REDQ | | |
| ENSEMBLE SIZE | NA | 10 |
| ENSEMBLE SUBSET SIZE | NA | 2 |
| DAC / SR-DAC | | |
| INITIAL OPTIMISM | $\beta_0^{ub}$ | 0.0 |
| INITIAL KL WEIGHT | $\tau_0$ | 1.0 |
| TARGET KL DIVERGENCE | $\mathcal{KL}^*$ | 0.5 |
| STANDARD DEVIATION MULTIPLIER | $f_\sigma$ | 1.25 |
| ADJUSTMENT LEARNING RATE | NA | $3e - 5$ |
| ADJUSTMENT LEARNING RATE BETA | NA | 0.5 |
| DAC+ | | |
| INITIAL STEPS | NA | 25000 |
| NETWORK SIZE | NA | $(256, 256, 256)$ |
| COPYING FREQUENCY | NA | every 20000 steps |

In the low replay ratio regime, all algorithms use 3 gradient steps per environment step. DAC performs parameter copying in 60000th and 500000th steps. In the high replay ratio regime,

all algorithms use 15 gradient steps per environment step. All algorithms perform full parameter resets in $[50000, 250000, 500000, 750000]$ steps. DAC performs parameter copying in $[25000, 125000, 375000, 625000, 875000]$ steps.

## F  TESTED ENVIRONMENTS

Below, we list tasks used in the low and high replay ratio regimes, as well as the ablation and hyperparameter studies. We choose tasks from the benchmark, but we drop those that are trivially solved by all the considered methods.

Table 2: 10 DMC tasks used in the model-free benchmark.

| DOMAIN | TASK |
|--------|------|
| CARTPOLE | SWINGUP-SPARSE |
| CHEETAH | RUN |
| FISH | SWIM |
| HUMANOID | RUN, STAND, WALK |
| HOPPER | HOP |
| PENDULUM | SWINGUP |
| SWIMMER | SWIMMER6 |
| QUADRUPED | RUN |

When comparing against model-based, we choose the tasks proposed by the authors of DreamerV3 (Hafner et al., 2023). As such, we use 18 DMC tasks of easy to moderate level of difficulty. Moreover, we use the DreamerV3 performance curves provided by the authors.

Table 3: 18 DMC tasks used in the DreamerV3 benchmark.

| DOMAIN | TASK |
|--------|------|
| ACROBOT | SWINGUP |
| CARTPOLE | BALANCE (SPARSE), SWINGUP (SPARSE) |
| CUP | CATCH |
| FINGER | SPIN, TURN-EASY, TURN-HARD |
| HOPPER | HOP, STAND |
| PENDULUM | SWINGUP |
| REACHER | EASY, HARD |
| WALKER | RUN, STAND, WALK |

## G  EXPERIMENTAL SETTINGS

### G.1  LOW AND HIGH REPLAY REGIMES

We consider the following set of continuous-action RL algorithms:

1. DAC / SR-DAC - Decoupled Actor-Critic. The approach proposed in this paper. DAC refers to the base algorithm; SR-DAC refers to DAC with a high replay ratio and periodical full-parameter resets.

2. SAC / SR-SAC - Soft Actor-Critic (Haarnoja et al., 2018) builds on DDPG (Silver et al., 2014). SAC extends the standard approach with the following: stochastic policy with gradients calculated via the reparametrization trick; automatic adjustment of entropy temperature; clipped Q-learning; and maximum entropy updates on actor and critic networks. SR-SAC refers to SAC with a high replay ratio and periodic resets of all networks (D'Oro et al., 2022).

3. OAC / SR-OAC - Optimistic Actor-Critic (Ciosek et al., 2019) extends SAC by designing an optimistic exploration policy that is a linear approximation to the Q-value upper bound.

OAC was shown to perform better than SAC in the regime of a low replay ratio. SR-OAC refers to OAC with a high replay ratio and periodic resets of all networks (D'Oro et al., 2022).

4. ND-TOP / SR-TOP - Non-Distributional Tactical Optimism and Pessimism (Moskovitz et al., 2021) builds. The algorithm builds on TD3 (Fujimoto et al., 2018) and addresses the optimism-pessimism problem by introducing an external discrete bandit that learns $\beta^{lb}$ from a set of predefined values. TOP was shown to perform better than OAC and SAC in the regime of a low replay ratio. SR-TOP refers to TOP with a high replay ratio and periodic resets of all networks (D'Oro et al., 2022).

5. RedQ - Randomized Ensembled Double Q-Learning (Chen et al., 2020) is an algorithm designed explicitly for high replay ratio regimes. RedQ avoids plasticity loss/overfitting by using an ensemble of Q-networks and performing every update using a small subset of the ensemble.

6. D4PG - Distributed Distributional Deterministic Policy Gradients (Barth-Maron et al., 2018) is a distributed version of DDPG leveraging distributional critics, prioritized experience buffer and N-step returns.

7. MPO - Maximum a Posteriori Policy Optimisation (Abdolmaleki et al., 2018) is an off-policy algorithm based on coordinate ascent on a relative entropy objective.

8. DreamerV3 - a state-of-the-art model-based algorithm (Hafner et al., 2023). It was shown to perform well across a wide range of tasks, including DMC. We use the results provided by the authors.

9. TD-MPC - Temporal Difference Learning for Model Predictive Control (Hansen et al., 2022) a state-of-the-art model-based algorithm leveraging model of the environment for critic-bootstrapped model-predictive control. In particular, TD-MPC was the first algorithm to reliably solve the dog domain tasks for proprioceptive control.

We take the maximum average performance of all algorithms (eg. first average over the seeds and tasks, then take the results from the $argmax$ of the average). In those experiments, the algorithms are tested on 10 tasks listed in Table 2. Each is run for $10^6$ environment steps and 10 random seeds.

## G.2 DreamerV3 and TD-MPC Comparison

In Figure 1d we compare DAC to DreamerV3. For this comparison, we use the results provided by the authors of Hafner et al. (2023) in their official repository. As such, we restrict the comparison to 18 tasks considered in the DreamerV3 paper (Hafner et al., 2023) listed in Table 3. For this comparison, the low replay DAC uses 10 seeds and DreamerV3 uses 5 seeds.

In Figures 7 and 12 we compare DAC to TD-MPC. For this comparison we use the results provided by the authors of Hansen et al. (2022) provided in their official repository. We run SR-SAC and SR-DAC for $10^6$ and SR-DAC+ for $30^6$ environment steps. For this comparison, SR-SAC and SR-DAC use 10 seeds each, whereas SR-DAC+ and TD-MPC use 5 seeds each.

## G.3 Ablation Studies

In Figure 4, we evaluate the impact of removing or adding components to DAC. The ablation study features DAC and 10 other variations thereof in low replay ratio (3 gradient steps per environment step) without using layer normalization on the critic network. Below, we describe the variations:

1. $(+KL)$ - in DAC, only the optimistic actor backpropagates the KL penalty portion of the loss (ie. only the optimistic actor adjusts its parameters such that the KL is minimized). We add the KL penalty to the conservative actor as well. This results in low divergence between the two actors, at the cost of both policies being suboptimal.

2. $(-\pi)$ - in this ablated setting, we consider SAC with only an optimistic actor. As such, this setting performs exploration, TD learning, and evaluation according to the optimistic policy. As in this version, there is only one actor, we cannot use the optimism adjustment mechanism.

3. $(-KL)$ - this tested configuration does not use KL penalty on any of the actors. As such, the optimistic actor is unconstrained and can diverge from the conservative one. This results in fully off-policy learning and poor coverage of the state-action space that is used in the TD updates.

4. $(+Det)$ - here, we set the conservative actor to be a fully deterministic policy. To this end, the optimistic actor performs exploration and is stochastic, whereas the conservative one performs deterministic TD updates. We find that such a setup quite often results in an overestimation of the Q-values, pinpointing towards regularizing properties of the soft SARSA. This is the first configuration to work better than the baseline SAC.

5. $(-\tau\&\beta)$ - this configuration does not use automatic adjustment of optimism or the KL penalty weight. As such, it is DAC with fixed optimism and fixed KL penalty weight throughout the training. Despite its simplicity, this configuration works significantly better than vanilla SAC.

6. $(-\beta)$ - this configuration does not use the automatic adjustment of optimism, but it still uses automatic adjustment of the KL penalty weight. Results indicate that the adjustment of optimism has a bigger impact on the performance than the adjustment of the KL penalty weight alone.

7. $(-\tau)$ - this configuration does not use automatic adjustment of KL penalty but still uses automatic adjustment of the optimism. Removing the automatic adjustment of the KL penalty did not have a very big impact on the performance. We hypothesize that this is due to the constant scale of returns between the environments.

8. $(-Copy)$ - here, we use DAC but we omit the optimistic actor parameter copying as well as optimism and KL penalty weight reinitialization. We find that parameter copying offered marginal performance benefits.

9. $(-\sigma)$ - in this version of the algorithm, we do not use the $\sigma$ multiplier. As such, the same level of variance is applied for exploration, as well as TD updates. We find that using $\sigma$ multiplier yields marginal improvements.

10. $(-LN)$ - here, we use complete DAC formulation, albeit we skip the layer normalization in the critic. We find that adding layer normalization increases the performance as compared to basic DAC.

We take the maximum average performance of all algorithms (eg. first average over the seeds and tasks, then take the results from the $argmax$ of the average). Each tested variation is run on the entire set of 10 hard DMC tasks listed in Table 2. Each variation is run for $500k$ environment steps and 10 random seeds. As such, the results shown in Figure 4 feature 1200 training runs.

### G.4    Hyperparameter Search

Figure 5 features the results of the hyperparameter stability tests. To this end, we conduct $500k$ environment step training runs of DAC with different hyperparameter settings in the low replay ratio regime (3 gradient steps per environment step). We consider a variety of values for 3 hyperparameters that are specific to DAC:

1. KL Divergence Target ($\mathcal{KL}^*$) - The target KL divergence between two actor networks. DAC adjusts optimism and KL penalty weight until the target divergence is met. We test 4 configurations of $\mathcal{KL}^*$.

2. Standard Deviation Multiplier ($f_\sigma$) - A multiplier defining how much bigger or smaller the standard deviation of an optimistic actor is as compared to the conservative one. We test 5 configurations of $f_\sigma$.

3. Copying Frequency - The frequency of optimistic actor copying conservative actor parameters. We consider the following settings: performing no copying; copying in the early stage of the training ($50k$ step); copying every $250k$ steps; copying every $125k$ steps and copying every $67.5k$ steps.

Similarly to the ablation study, each hyperparameter setting is tested on tasks listed in Table 1. Each setting is run for $500k$ environment steps and 10 random seeds. As such, the results shown in Figure

5 feature 1300 training runs. We find that DAC consistently improves on SAC performance across a variety of hyperparameter setting.

## H    LEARNING CURVES

In the following section, we share the training curves for the Dreamer, low replay ratio, and high replay ratio experiments.

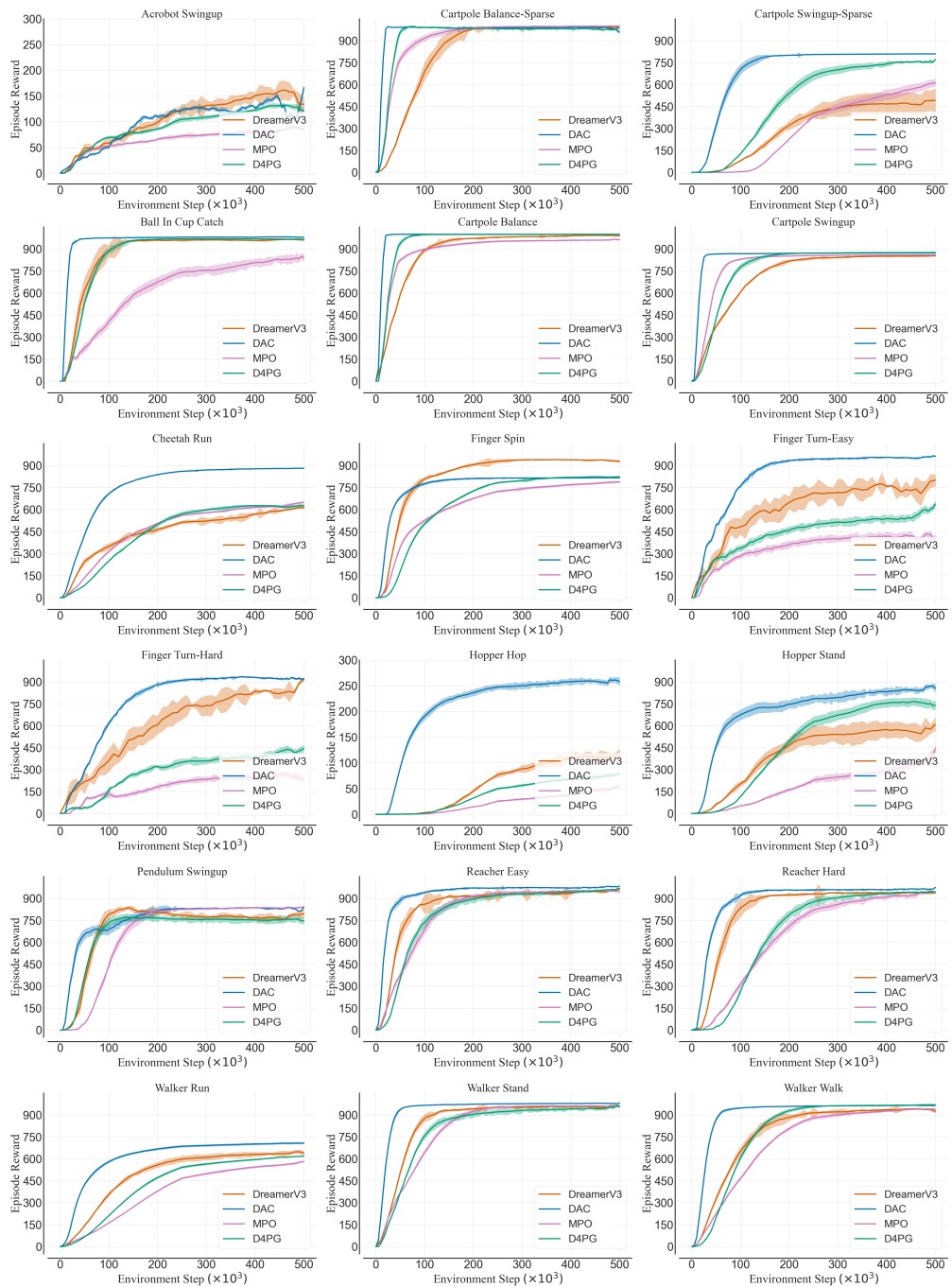

Figure 13: Results for the Dreamer evaluation. DAC uses no LayerNorm, a low replay ratio ($RR = 3$) and no parameter resets. $500k$ steps, 10 seeds, mean and 3 standard deviations CI.

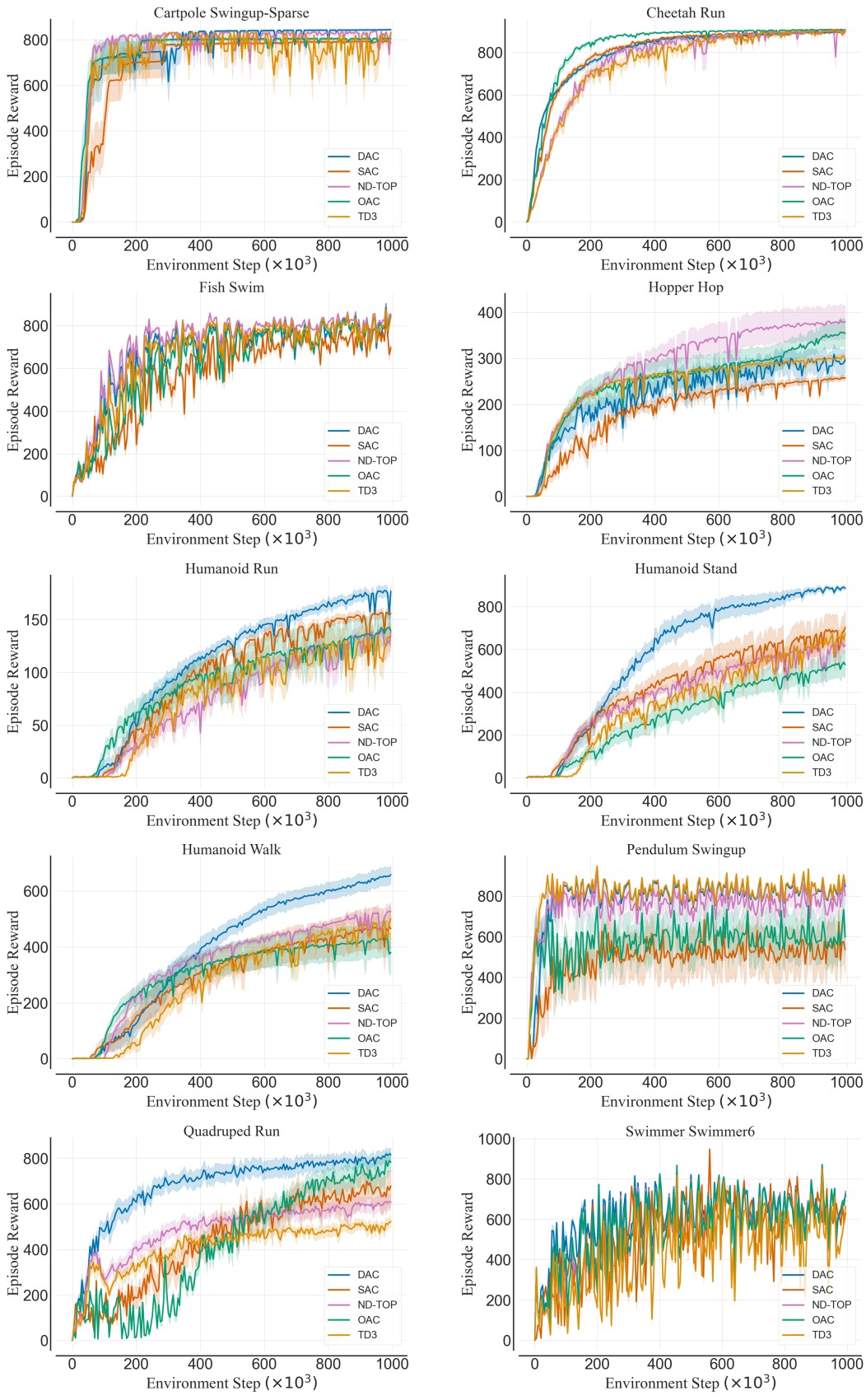

Figure 14: Results for low replay ratio ($RR = 3$) experiments. $10^6$ environment steps, 10 seeds, mean, and 3 standard deviations CI.

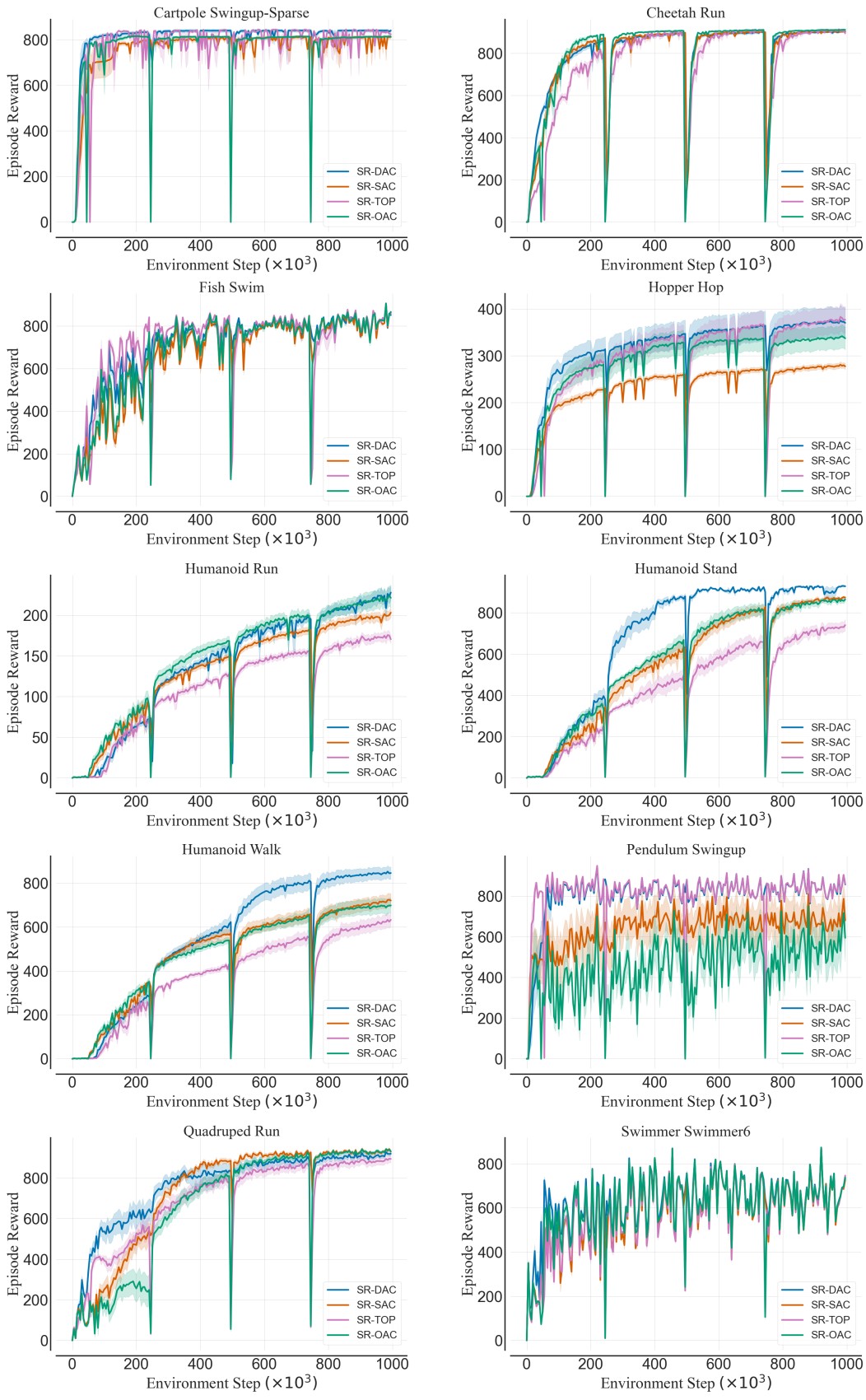

Figure 15: Results for high replay ratio ($RR = 15$) experiments. $10^6$ environment steps, 10 seeds, and 3 standard deviations of the mean.

