# OpenReview forum: "Decoupled Actor-Critic"
_ICLR.cc/2024/Conference — Submitted to ICLR 2024_

### Official Review · Reviewer_HF8k · 2023-10-29

**Soundness:** 2 fair
**Presentation:** 1 poor
**Contribution:** 2 fair
**Rating:** 5
**Confidence:** 4

**Summary:**

This is an empirical paper well-motivated by solving the conservative policy update and optimistic exploration problem that exists in deep RL. The proposed method achieves significant performance compared with the presented baselines.

**Strengths:**

The motivation is clear, and the method looks solid.
The authors provide extensive experiments and put great effort into enhancing reproducibility.

**Weaknesses:**

The paper does not compare against the ensemble-based methods [e.g., REDQ], which I believe is relevant and necessary.

On the general applicability of the idea, will this idea work with different value-to-policy generation rules?

There is no text pointing to Figure 1 and Figure 2. Also, many of the abbreviations are used since the beginning of the paper but are introduced at very late stages. The overall presentation of the paper can be improved.

The authors do not disclose any pitfalls of the DAC algorithm. Is it consistently better than SAC/TD3/other baselines? This is a huge claim, and would definitely be a huge strength of the work if it is true.





References:

[REDQ] Chen, Xinyue, et al. "Randomized ensembled double q-learning: Learning fast without a model." arXiv preprint arXiv:2101.05982 (2021).

**Questions:**

It is known in the literature that off-policy learning can suffer from the [tandem problem]. How do you solve such difficulty when updating the conservative actor using the data generated by the optimistic actor?

What is the backbone of DAC? How do the authors explicitly model the variance and mean of the policy?


References:

[Tandem] Ostrovski, Georg, Pablo Samuel Castro, and Will Dabney. "The difficulty of passive learning in deep reinforcement learning." Advances in Neural Information Processing Systems 34 (2021): 23283-23295.

---

> ### Author Response · Authors · 2023-11-14
> **Response to Reviewer HF8k (1)**
>
> We thank the Reviewer for their time writing the review. We are happy that the Reviewer finds the proposed approach “well-motivated” and “solid”, the performance gains “significant” and the experiments “extensive”. We would like to point the Reviewers attention to new experimental results described in our meta-comment. The new results further indicate that DAC achieves very strong performance on locomotion tasks, significantly outperforming both model-based (TD-MPC) and model-free (SR-SAC) SOTA on the dog domain. Below, we respond to each question/weakness listed by the Reviewer.
>
> > The paper does not compare against the ensemble-based methods [e.g., REDQ], which I believe is relevant and necessary.
>
> We kindly ask the Reviewer to investigate Figures 1, 9 & 10 of the original manuscripts to find the evaluation of DAC against exactly REDQ. As follows, REDQ achieves significantly worse results than other considered algorithms in the high replay benchmark. This result is consistent with [1], where SR-SAC (high replay SAC with resets) massively outperforms REDQ across various replay regimes. For completeness, we included both REDQ and SR-SAC in the evaluation presented in Figure 1, but dropped it in most of the latter evaluations because of substantial performance differences between REDQ and other algorithms.
>
> > On the general applicability of the idea, will this idea work with different value-to-policy generation rules?
>
> Our approach is trivially applicable to other value-based continuous actions algorithms like DDPG, TD3 or Dreamer. It would require some accommodations to be applied to discrete actions algorithms like Rainbow or BBF. We are happy to expand on the answer if the Reviewer shares which particular algorithm the Reviewer is interested in.
>
> > There is no text pointing to Figure 1 and Figure 2. Also, many of the abbreviations are used since the beginning of the paper but are introduced at very late stages. The overall presentation of the paper can be improved.
>
> We thank the Reviewer for this suggestion! We added text in the main body that describes why Figures 1 & 2 are important for our narrative. We make some additional changes to make sure that abbreviations are explained earlier in the text.
>
> > The authors do not disclose any pitfalls of the DAC algorithm.
>
> We ask the Reviewer to note that our original manuscript contains a “Limitations” section, where we describe pitfalls of our proposed DAC algorithm. Those limitations include:
>
> 1. Complexities of the tandem setting - “DAC divergence minimization presents unique optimization challenges. Unlike typical uses of KL divergence, where the target distribution remains fixed (eg. Variational Autoencoders (VAE)), DAC deals with a constantly evolving policy that is continually improving. Consequently, the optimistic actor needs to keep up with the conservative actor’s changes. As depicted in Figure 4, DAC heavily relies on maintaining a low divergence between the actors. While DAC adjustment mechanisms proved effective in the tested environments, there is no guarantee that they will suffice in more complex ones.”
> 2. Optimistic actor architecture - “The second drawback of DAC lies in its inherent use of two actor networks, which results in slightly increased memory and computational demands compared to the standard Soft Actor-Critic (SAC) approach. In practice, the wall-clock time of DAC is around 10% greater than that of SAC and is indistinguishable from the overhead induced by OAC, which requires additional backpropagation through the critic ensemble. Moreover, since DAC initializes both actors with identical parameters, they must share the same network architecture. However, as indicated by Figure 4, simply copying parameters between them offers only minimal performance enhancement. In light of this, we believe that the necessity for identical architectures can be mitigated by employing techniques like delayed policy updates or by learning rate scheduling.”
>
> As DAC builds on SAC, we focus on limitations stemming from components specific to DAC. We add a line in the limitations section mentioning that DAC inherits limitations of SAC.

---

> > ### Author Response · Authors · 2023-11-14
> > **Response to Reviewer HF8k (2)**
> >
> > > Is it consistently better than SAC/TD3/other baselines? This is a huge claim, and would definitely be a huge strength of the work if it is true.
> >
> > As shown in our experiments (Figures 1, 4, 5 & 6), DAC indeed achieves significantly better performance than other baselines within our experiments regime (DMC locomotion). Below, we leave additional comments for each of the Figures:
> >
> > 1. Figure 1 - Here, we show sample efficiency (mean performance across all tested tasks) for two replay ratio regimes. The Figure contains a 95% CI calculated using RLiable package [2]. In both regimes, there is a significant gap between DAC and other considered baselines (SAC, TD3, OAC, TOP-TD3, DreamerV3, REDQ, SR-OAC, SR-TOP, SR-SAC (current SOTA on DMC locomotion)).
> > 2. Figure 4 - Here, we evaluate the impact of every design decision on DAC performance and compare such simplified DAC algorithms to SAC. We show that DAC achieves better performance than SAC/TD3, even after many simplifications to the design.
> > 3. Figure 5 - Here, we evaluate the impact of DAC hyperparameters on its performance. In our manuscript, the Figure contains a reference SAC performance, showing that DAC significantly outperforms SAC for all tested hyperparameter settings.
> > 4. Figure 6 - Here, we show final performance statistics generated using RLiable package [2]. DAC achieves significantly better performance than the considered baselines (SAC, TD3, OAC, TOP-TD3, DreamerV3, REDQ, SR-OAC, SR-TOP, SR-SAC (current SOTA on DMC locomotion).
> >
> > All evaluations are conducted using 10 tasks and 10 seeds per task. We note that the results described above constitute more than 2000 training runs. Considering the documented results, we write in our abstract: “DAC achieves state-of-the-art performance and sample efficiency on locomotion tasks”. We also note that the newly provided experiments are consistent with this assessment. We restrict the claim to locomotion tasks, as those types of tasks evaluated in our work. We agree that this is a big claim. However, even if DAC in presented design doesn’t instantaneously transfer to different environments than prio-based DMC, we believe the experimental results that indicate significant improvements on the widely used DMC benchmark are of interest to the RL community. Furthermore, new experiments on the dog domain indicate that DAC can solve tasks entirely unsolved by SAC (dog-walk, dog-trot, dog-run)
> >
> > > It is known in the literature that off-policy learning can suffer from the [tandem problem]. How do you solve such difficulty when updating the conservative actor using the data generated by the optimistic actor?
> >
> > We kindly ask the Reviewer to note that Section 3.2 of our original manuscript describes the KL penalty term that ensures that the two policies do not diverge substantially. In this section we write: “KL divergence, the second objective term, regularizes the optimistic policy. (...) Using KL stabilizes the off-policy learning by ensuring that the sampled trajectories are probable under the conservative actor policy.” Furthermore, in Section A2 we write: “the optimistic policy objective features a KL divergence penalty. As shown in ablation studies in Figure 4, such a penalty is of crucial importance for DAC performance. The penalty is applied only to the optimistic actor and serves multiple purposes: (...)”. To this end, the KL penalty term is introduced specifically to restrict the negative effects of the tandem setting. Furthermore, we employ hard parameter copying as a final fail-safe mechanism against divergence of two actors. As such, DAC goes a long way to restrict the negative effects of the tandem setting on learning.
> >
> > > What is the backbone of DAC? How do the authors explicitly model the variance and mean of the policy?
> >
> > DAC is just SAC with an additional “optimistic” actor network (which is updated according to a rule described in Section 3.2 & 3.3). In Section 3 we write: “DAC introduces two distinct actor networks: an optimistic one and a conservative one. (...) While policies can be represented by various parameterized distributions, we implement both actors as simple diagonal normal distributions, transformed by the hyperbolic tangent activation.” Additionally, Section A.2 goes in detail about why we used soft learning paired with hyperbolic tangent transformed policies.
> >
> > We thank the Reviewer for their work and comments. We hope that the above responses and changes to the manuscripts clear the Reviewers doubts about our proposed approach. We kindly ask the Reviewer to share which parts of our work remain unclear or to adjust the score of the manuscript accordingly.
> >
> > [1] D'Oro, Pierluca, et al. "Sample-Efficient Reinforcement Learning by Breaking the Replay Ratio Barrier." The Eleventh International Conference on Learning Representations. 2022
> >
> > [2] Agarwal, Rishabh, et al. "Deep reinforcement learning at the edge of the statistical precipice." Advances in Neural Information Processing Systems. 2021

---

> > > ### Author Response · Authors · 2023-11-22
> > > **Further response to Reviewer HF8k**
> > >
> > > The Reviewer brought 6 weaknesses/questions in their review, out of which 4 were addressed in the originally reviewed manuscript and the remaining 2 were comprehensively answered in our rebuttals. As the rebuttal period is closing in few hours, we kindly ask the Reviewer to acknowledge our rebuttal and reconsider their initial score.

---

### Official Review · Reviewer_7ruN · 2023-10-31

**Soundness:** 3 good
**Presentation:** 2 fair
**Contribution:** 2 fair
**Rating:** 6
**Confidence:** 3

**Summary:**

The optimistic have different impact on actor and critic update, i.e., the overestimation of critic may result in sub-optimal policy while the optimistic policy can yield lower regret. Thus motivated, this work propose to decouple the actor-critic by using different actors for TD learning and exploration. The proposed method is tested on locomotion tasks with various replay ratio and achieve better performance than previous work.

**Strengths:**

1. This paper is well-motivated by the issues in the single policy regime in the conventional actor-critic.
2. The proposed method shows less sensitivity to the hyperparameter thanks to the adaptive adjustment of the optimistic level, which is different from the related work Optimistic Actor-Critic (OAC).

**Weaknesses:**

See the questions below

**Questions:**

1. It is unclear how does Figure 2b and 2c shows the critic disagreement for different actors?
2. If it is possible to use non-linear approximation of Q-value in Eqn. (6)
3. What is the impact of the ensemble size as in Eqn. (4)
4. Some notations can be confusing, i.e., $|\mathcal{A}|$ is used to denote the action dimensionality where normally it is used as the cardinality of the action space.

---

> ### Author Response · Authors · 2023-11-14
> **Response to Reviewer 7ruN**
>
> We thank the Reviewer for their time writing the review. We are happy that the Reviewer finds our approach “well-motivated” and appreciated the proposed adaptive mechanisms that address OAC shortcomings. We would like to point the Reviewer’s attention to the new results added to our experiments (described in our meta-comment). In those results, DAC appears to significantly outperform model-based DMC SOTA (TD-MPC) and to the best of our knowledge represents the best recorded sample efficiency on the dog/humanoid domains of a model-free RL algorithm . Please find our responses to the Reviewer’s questions below.
>
> > It is unclear how does Figure 2b and 2c shows the critic disagreement for different actors?
>
> Figure 2b & 2c show conservative and optimistic policies and related sampled state-action samples (denoted by different dot colors - red for optimistic and black for conservative). As follows, the policies are not massively different which is due to a small KL divergence between the two policies which is enforced by DAC. Despite similarity of the policies, the optimistic policy allocates some of the allowed KL budget to achieve bigger variance than the baseline conservative policy (indicated by the smaller likelihoods of the optimistic policy); secondly, the optimistic policy leads the agent to the corners of the state-action space which as shown in Figure 2a yields most critic disagreement. If the Reviewer wishes, we are happy to create a new graph where the optimistic policy is less penalized for the divergence and should thus lead to more contrasting policies.
>
> > If it is possible to use non-linear approximation of Q-value in Eqn. (6)
>
> Eqn. 6 describes the optimistic policy rule used in OAC (not in our proposed DAC), which as described by the authors of OAC used a linear approximation. To the best of our knowledge, some of the closed-form solutions used in OAC require the model to be linear. In fact, one of the key novelties of DAC that contrast to previous work is that it uses a non-linear model of Q-value upper-bound.
>
> > What is the impact of the ensemble size as in Eqn. (4)
>
> With more than two critics, the minimum of the ensemble stops being equivalent to Eqn. 5 with $\beta^{lb} = -1$. However, it can be easily calculated what $\beta^{lb}$ corresponds to a clipped double Q-learning rule with more than two critics (this is discussed in [1]). We believe that extending DAC to a bigger ensemble is an interesting direction. Approaches like REDQ take a subset of the ensemble for every TD calculation and we think that this is a design choice that would have to be thoroughly ablated in the context of DAC. As such, we believe that it mandates an independent research endeavor.
>
> > Some notations can be confusing, i.e., |A| is used to denote the action dimensionality where normally it is used as the cardinality of the action space.
>
> We used the notation inherited from the SAC manuscript, which uses |A| to denote control dimensions. We though that such notation is fitting, as we consider control over continuous space. We are happy to change the notation according to the Reviewer's proposition.
>
> We hope that the above responses anwer the Reviewers' doubts about our proposed method. Furthermore, we hope that the new experimental results presenting DAC performance on complex DMC environments, as well as changes done to the manuscript increase the Reviewers confidence in DAC. If so, we kindly ask the Reviewer to consider adjusting the score of our work.
>
> [1] - Ciosek, Kamil, et al. "Better exploration with optimistic actor critic." Advances in Neural Information Processing Systems. 2019.

---

> > ### Comment · Reviewer_7ruN · 2023-11-21
> > **Thank the authors for the response**
> >
> > I appreciate the authors addressing my concerns and I will keep my original score.

---

> > > ### Author Response · Authors · 2023-11-22
> > > **We thank the Reviewer for their suggestions**
> > >
> > > We thank the Reviewer for their time and for the appreciation of our contribution. We are happy that our rebuttal has addressed all of the Reviewers concerns.
> > >
> > > If possible, we kindly ask the Reviewer for opinions on how to further improve the quality of the manuscript - we are dedicated to implement further changes if needed.

---

### Official Review · Reviewer_XQ6P · 2023-10-31

**Soundness:** 3 good
**Presentation:** 3 good
**Contribution:** 3 good
**Rating:** 6
**Confidence:** 4

**Summary:**

This paper proposes the Decoupled Actor-Critic (DAC) algorithm that leverages two actors, one optimistic for efficient environment exploration and one conservative for stable learning. DAC further features a adaptive mechanism for setting the optimism trade-off to better account for the impact of reward scales. The performance of DAC is evaluated on a variety of tasks from the DeepMind Control Suite and compares favorably to the selected baselines.

**Strengths:**

-	The approach of combining optimistic exploration with conservative updating is very neat
-	Evaluation on 10 seeds with a multitude of baselines is great
-	Overall well written / structured paper that is easy to follow
-	Promising results, while some adjustments should be made regarding the experimental evaluation as discussed below

**Weaknesses:**

-	Dreamer results in Figure 11 are looking good, while harder tasks such as Quadruped Run and Humanoid Walk are missing – Dreamer-v2 is able to solve these tasks well even for visual-control, why not compare on tasks like Quadruped/Humanoid/etc. (or even extend to visual control)?
-	More complex Control Suite tasks (Figures 12 & 13) like Humanoid Walk/Run should be run for longer as they have not converged, yet, while recent papers have also evaluated on the Dog domain
-	SAC is in general a good baseline, however, it would be nice to also compare performance to a more “DMC-native” baseline such as D4PG or (D)MPO to provide another reference point
-	There are quite a few missing articles / words + typos that should be fixed
-	It would be good to extend the discussion to model-based exploration agents, e.g. the works in [1] and [2] leveraged Dreamer/RSSM-based agents for visual control that explored via ensemble disagreement over rewards, where [1] maintained an optimistic upper confidence bound exploration policy as well as a distinct exploitation policy. [3] also explores uncertain returns with access to the nominal reward functions.

[1] T. Seyde, et al. „Learning to plan optimistically: Uncertainty-guided deep exploration via latent model ensembles”, CoRL 2021.

[2] R. Sekar, et al. “Planning to explore via self-supervised world models,” ICML 2020.

[3] P. Ball, et al. "Ready policy one: World building through active learning," ICML, 2020.

**Questions:**

-	Could your provide an exemplary calculation of how the maximum average performance is calculated in Section F.1 – is the argmax over time? Why average over the tasks?
-	Fish and Swimmer are very stochastic tasks due to random goal placement, but the results in Figure 12 & 13 still look extremely variable with confidence intervals barely visible. Could you double check how the evaluations are computed? Do all runs use the same evaluation seed (e.g. same eval goal across all seeds)?
-	It might be worth briefly discussing the general impact of replay ratios across algorithm implementation, as the “low replay” regime with 3 gradient steps per 1 environment steps seems to be significantly higher than e.g., Acme’s MPO default of 1 gradient steps per 8 environment steps (or 1/1 for Acme’s SAC). The impact of replay ratios can be wild.
-	The caption of Table 2 mentions 10 HARD DMC tasks, while Pendulum Swingup and Cartpole Swingup Sparse would not be considered hard (arguably, Cheetah, Quadruped, Fish aren’t hard either)?
-	Have you also tried DroQ as an even more recent addition / alternative to REDQ?
-	Have you tried an ablation study on the number of critics? What patterns would you expect?

---

> ### Author Response · Authors · 2023-11-14
> **Response to Reviewer XQ6P (1)**
>
> We thank the Reviewer for a thorough and insightful review. We are excited that the Reviewer finds our approach “neat”, the paper “well written” and the results “promising”. We extended our experiments by a setting suggested by the Reviewer - 3mln steps on humanoid and dog tasks. Furthermore, we added an experiment verifying our claim regarding overestimation and plan on number of changes in the manuscript. We are excited to share that the new results further indicate that DAC is a substantial improvement over the existing model-free approaches. We described the new additions in detail in our meta-response. Below, we directly respond to the Reviewers’ questions.
>
> > Dreamer results in Figure 11 are looking good, while harder tasks such as Quadruped Run and Humanoid Walk are missing – Dreamer-v2 is able to solve these tasks well even for visual-control, why not compare on tasks like Quadruped/Humanoid/etc.?
>
> We used the results provided by the authors of DreamerV3 in their official repository. As such, we restricted evaluation to the 18 DMC tasks considered in DreamerV3 prio evaluation. DreamerV3 has pretty substantial compute requirements as compared to model-free algorithms. To that end, we are unsure if the potential small changes to the evaluation (there are already 18 tasks in the benchmark) mandate the cost of running Dreamer on additional environments.
>
> > (or even extend to visual control)?
>
> We agree that prio-based control is not where Dreamer shines. However, we also think that using DAC in visual tasks would require non-trivial architectural choices. Such non-trivial choices stem from the intersection of using two actor networks, the common practice in visual control to use a shared encoder for actor-critic structure, and the recently researched impact of gradient weighting between actor-critic in visual tasks [1]. As such, we would have to ablate on various design choices regarding architecture and gradient passing in our extended actor-critic context. As such, we think that extending DAC to visual control validates a separate research endeavor. We kindly ask the Reviewer to note that it is a common practice to evaluate a continuous-control RL algorithm on a prio-based benchmark alone [2, 3, 4].
>
> > More complex Control Suite tasks (Figures 12 & 13) like Humanoid Walk/Run should be run for longer as they have not converged yet, while recent papers have also evaluated on the Dog domain
>
> DMC 500k/1mln has become a popular evaluation protocol used in recent works [1,5,6]. We think that running many tasks of varying difficulty for a consistent amount of timesteps makes sense from the aggregate evaluation perspective. However, we also agree that harder, unsolved tasks are more interesting than well researched propositions like cheetah. To this end, we added a new experiment: dog {walk, run, trot} and humanoid {walk, run} on 3mln environment steps - following the Reviewer’s expectations. We describe the experiment in our meta-response. There, we compare SR-SAC, DAC and a recent baseline pointed by the Reviewer (TD-MPC). Please find a summary table below:
>
> |    task   | SR-SAC |  DAC |   DAC+  | TD-MPC | DAC+ | TD-MPC |
> |:---------:|:------:|:----:|:-------:|:------:|:----:|:------:|
> |    mean   |   214  |  439 | **607** |   546  |  **756** |   725  |
> | env steps |  1mln  | 1mln |   1mln  |  1mln  | 3mln |  3mln  |
>
> In particular, SR-SAC does not seem to work on dog domain. This result is consistent with the evaluation performed in the TD-MPC manuscript.
>
> > SAC is in general a good baseline, however, it would be nice to also compare performance to a more “DMC-native” baseline such as D4PG or (D)MPO to provide another reference point
>
> We are happy to share that we added D4PG and MPO to the Dreamer evaluation. The results are consistent with the Dreamer manuscript (ie. DAC > DreamerV3 > D4PG > MPO)
>
> > There are quite a few missing articles / words + typos that should be fixed
>
> We made a sweep through the manuscript, we will make sure that the camera-ready version is fixed!
>
> > It would be good to extend the discussion to model-based exploration agents (...)
>
> Thank you for pointing us to those works! We have added a paragraph to the Related Work section detailing how DAC related to this literature.

---

> > ### Author Response · Authors · 2023-11-14
> > **Response to Reviewer XQ6P (2)**
> >
> > > Could your provide an exemplary calculation of how the maximum average performance is calculated in Section F.1 – is the argmax over time? Why average over the tasks?
> >
> > Of course, we leave an example below:
> >
> > RESULT is an array of shape TIMESTEPS x SEEDS x TASKS. We select a single timestep t and report results for all tasks and seeds for this particular timestep t.  Using the NumPy convention such timestep t is calculated via: RESULT.mean(axis=2).mean(axis=1).argmax(axis=0). Thanks to this pattern, there is an exact relation between the sample efficiency graphs (Figure 1) and the aggregate results (Figure 6) - results in Figure 6 correspond to maximum values presented in Figure 1. An alternative popular approach is to take an average performance in the last K timesteps, which would be calculated according to: RESULT[-K:].mean(). We calculate the aggregate percentage DAC improvement over SAC using both patterns in the table below:
> >
> > |  Experiment Regimne |    Method 1 (used in the paper)   |  Method 2 (second example; K=10)  |
> > |:-------------------:|:---------------------------------:|:---------------------------------:|
> > |        RR = 3       | 20% (IQM) 18% (Mean) 21% (Median) | 19% (IQM) 18% (Mean) 25% (Median) |
> > |       RR = 15       |   7% (IQM) 7% (Mean) 5% (Median)  |  6% (IQM) 7% (Mean) 11% (Median)  |
> > | Average Improvement |                13%                |                14%                |
> >
> > We note that the two protocols result in statistically negligible differences. We are happy to change it to whichever pattern the Reviewer finds less confusing for the reader.
> >
> > > Fish and Swimmer are very stochastic tasks due to random goal placement, but the results in Figure 12 & 13 still look extremely variable with confidence intervals barely visible. Could you double check how the evaluations are computed? Do all runs use the same evaluation seed (e.g. same eval goal across all seeds)?
> >
> > We built our implementations on the codebase associated with [6] and did not change the environment wrappers in any way. We think that the variability of our results is consistent with [6], considering the differences in the evaluation protocol (10 vs. 5 seeds; 3 std vs. 95% CI)
> >
> > > It might be worth briefly discussing the general impact of replay ratios across algorithm implementation, as the “low replay” regime with 3 gradient steps per 1 environment steps seems to be significantly higher than e.g., Acme’s MPO default of 1 gradient steps per 8 environment steps (or 1/1 for Acme’s SAC). The impact of replay ratios can be wild.
> >
> > We are happy to add an appendix section discussing the impact of replay ratio on RL algorithms. Indeed, we believe many approaches could be revisited, given that (replay ratio + resets) is applicable to almost any off-policy algorithm. Unfortunately, some robot hardware might impose inherent restrictions to the replay ratio. As such, we believe researching new algorithms in both high and low replay settings is valuable. Furthermore, we would like to note that the baselines ran within our experiments (ie. Figure 6) use consistent replay ratio and reset schedules.
> >
> > > The caption of Table 2 mentions 10 HARD DMC tasks, while Pendulum Swingup and Cartpole Swingup Sparse would not be considered hard (arguably, Cheetah, Quadruped, Fish aren’t hard either)?
> >
> > We thank the Reviewer for pointing this mistake out. Indeed, the model-free evaluation considers tasks from easy to hard levels of difficulty. We change the Table 2 caption in the manuscript.
> >
> > > Have you also tried DroQ as an even more recent addition / alternative to REDQ?
> >
> > Unfortunately we haven’t tried DroQ. We used the JaxRL library which features only an implementation of REDQ.
> >
> > > Have you tried an ablation study on the number of critics? What patterns would you expect?
> >
> > Increasing the critic ensemble size in DAC is an interesting research theme. Approaches like REDQ take a subset of the ensemble for every TD calculation. We think that such design choice would have to be ablated in the context of DAC. To cut to the chase, we believe that ensemble-based DAC is an interesting direction due to the synergies specific to DAC which are absent in standard SAC (ie. learning based on the ensemble statistics). Since enlarging the critic ensemble would both imply substantially higher compute requirements and distort the evaluation against baselines we decided to leave it for future work.
> >
> > We thank the Reviewer for their insightful questions and comments, which were greatly helpful in enhancing the quality of the manuscript and experiments. We hope that the Reviewer finds new changes convincing and will adjust the score accordingly.

---

> > > ### Author Response · Authors · 2023-11-14
> > > **Response to Reviewer XQ6P (3)**
> > >
> > > [1] Yarats, Denis, et al. "Improving sample efficiency in model-free reinforcement learning from images." Proceedings of the AAAI Conference on Artificial Intelligence. 2021.
> > >
> > > [2] Haarnoja, Tuomas, et al. "Soft actor-critic: Off-policy maximum entropy deep reinforcement learning with a stochastic actor." International Conference on Machine Learning. 2018.
> > >
> > > [3] Ciosek, Kamil, et al. "Better exploration with optimistic actor critic." Advances in Neural Information Processing Systems. 2019.
> > >
> > > [4] Chen, Xinyue, et al. "Randomized Ensembled Double Q-Learning: Learning Fast Without a Model." International Conference on Learning Representations. 2020.
> > >
> > > [5] D'Oro, Pierluca, et al. "Sample-Efficient Reinforcement Learning by Breaking the Replay Ratio Barrier." The Eleventh International Conference on Learning Representations. 2022.
> > >
> > > [6] Nikishin, Evgenii, et al. "The primacy bias in deep reinforcement learning." International Conference on Machine Learning. 2022.

---

> > > > ### Comment · Reviewer_XQ6P · 2023-11-20
> > > > **Response to Rebuttal**
> > > >
> > > > Thank you for your extensive replies. I think the additional experiments and explanations improved the paper and I'm happy to increase my rating. Regarding the computation of aggregate scores, it might make sense to add a brief variation of the above explanation to the manuscript for ease of readability.

---

> > > > > ### Author Response · Authors · 2023-11-22
> > > > > **We thank the Reviewer for improvements to our manuscript**
> > > > >
> > > > > We thank the Reviewer for their input and suggested changes to the manuscripts. We are happy that the new material meets the Reviewer’s expectations. Per Reviewer suggestions, we will add further explanation on how we aggregate our results.
> > > > >
> > > > > As of now, the Reviewer indicates that our contribution is “marginally above the acceptance threshold” - we thank the Reviewer for the appreciation of our work. As we strive to deliver the best possible material to the community, we kindly ask the Reviewer for suggestions of content that the Reviewer expects out of work that the Reviewer would call a “good paper”. In case of the acceptance of our manuscript, we are happy to execute the suggestions.

---

### Official Review · Reviewer_SCgY · 2023-10-31

**Soundness:** 2 fair
**Presentation:** 3 good
**Contribution:** 2 fair
**Rating:** 6
**Confidence:** 3

**Summary:**

The paper claims that there seems to be a conflicting demand in actor-critic architecture: the critic tends to overestimate so conservatism is needed when computing bootstrap target; however, the actor should act optimistically to improve sample efficiency/reduce regret. The authors propose dual actor-critic to reconcile the problem: there are both conservative actor and optimistic actor, and an ensemble of critics used to compute the lower/upper bound of value estimates. The basic idea is to let actor acts optimistically while the other actor act is used for maximize lower bound of the Q values. The algorithm also adds some heuristic designs such as minimizing the KL divergence between the two actors, learning the optimism control parameter and the KL divergence weight. Empirical results are provided to show the effectiveness of the algorithm.

**Strengths:**

1. The paper includes many experiments, which might provide heuristics for application-oriented tasks;

2. The paper presents its algorithm clearly.

3. The studied problem regarding balancing optimism and potential overestimation is interesting.

**Weaknesses:**

The proposed algorithm is mostly designed by heuristics and the implementation details are not theoretically justified. Although I do not think theoretical support is necessary for a good paper, I expect empirical evidence to verify the critical claims/algorithmic designs of this paper (see below).

Furthermore, since the proposed algorithm is basically a synthesis of different intuitive designs, a discussion of where the algorithm would converge to should be provided. Currently the paper is written in a way that different updating rules are introduced; I expect to see a clear objective function (maybe with constraints) of Algorithm 1, so readers can easily see what it is optimizing, can the authors write it down in the rebuttal?

When simultaneously maximizing both lower and upper bound of the Q values, would it squash all action values higher and still result in overestimation?

empirically:
1. Verify the proposed method indeed mitigate overestimation comparing with an algorithm without using any correction, e.g., compare the estimate value and MC estimation; the current version of the paper directly using evaluation return as a performance measure, which, I think lacks justification, as it is unclear where the improved performance results from;

2. The added optimism is essentially an exploration strategy, some baselines aiming at exploration should be also compared.

3. Ablation study should be provided to justify the following design choices: the effect of ensemble (and do those baselines use ensemble too?), the necessity of optimizing optimism and KL weight (can you use some intuitive choices instead of learning them)?

Any comments how do you decide the order of the updating rules 10-15? And how their learning rates are chosen?

4. The algorithm introduces many more hyper parameters comparing with commonly seen SAC or TD3 due to the added components in the losses, I would not consider the comparison to be fair with other baselines, unless evidence of similar efforts have been made to thoroughly sweep baseline’s hyper-parameters is provided.

5. Eq 7, how do you calculate the gradient w.r.t optimistic actor parameters, it appears the gradient should be also propagated through/to the first two Q functions and these Qs are interdependent with actor.

6. The motivation of the paper is to avoid overestimation while keep good exploration. Isn’t it quite intuitive to combine some exploration method with methods that mitigate overestimation?  it shouldn’t be difficult to design such a baseline as mitigating overestimation typically require multiple critics and uncertainty estimate could be derived from the ensemble for exploration purpose. What is the proposed algorithm’s advantage?

7. The experimental design of studying replay ratio appears to disconnect with the primary motivation of this paper; this should be put in the appendix.

**Questions:**

see above.

---

> ### Author Response · Authors · 2023-11-14
> **Response to Reviewer SCgY (1)**
>
> We thank the Reviewer for their time. We are happy that the Reviewer finds our algorithm presentation clear, the considered problem interesting and that the Reviewer appreciates the amount of experiments performed within our work. We particularly thank the Reviewer for the suggestion regarding the overestimation experiment. Furthermore, we perform additional experiments on the dog domain. There, we record that DAC achieves significantly better performance than model-based and model-free DMC SOTA. We describe the experiments in our meta-response. Below, we respond in detail to each of the Reviewer’s questions.
>
> > The paper claims that there seems to be a conflicting demand in actor-critic architecture
>
> This conflict is not something that we discuss first - it has been thoroughly discussed in [1, 2]. We hope that our writing does not imply otherwise.
>
> > The proposed algorithm is mostly designed by heuristics and the implementation details are not theoretically justified.
>
> We would like to note that DAC follows fundamental concepts in RL theory, such as the actor-critic framework (stemming from the Policy Gradient Theorem), optimism in the face of uncertainty and off-policy learning. Its design incorporates widely researched principles of policy and value iteration in the context of maximum entropy formulation, which all are foundational for the field of RL. Even the automatic optimization of optimism and KL regularization can easily be linked to dual optimization [3], used in similar contexts in many other RL algorithms [4, 5]. However, we realize that our manuscript focuses on evaluating implementation of theoretically founded principles in practice, and as such could be considered an empirical work.
>
> >  I expect empirical evidence to verify the critical claims/algorithmic designs of this paper
>
> Figures 4 & 5 extensively evaluate the performance impact of many design decisions (10 DAC simplifications & 14 hyperparameter configurations). We note that 7/10 DAC simplifications and 14/14 hyperparameter configurations perform better than baseline SAC.
>
> > Furthermore, since the proposed algorithm is basically a synthesis of different intuitive designs, a discussion of where the algorithm would converge to should be provided.
>
> In Section 3.2 we write: “This mechanism allows for separate entropy for TD learning and exploration while retaining standard convergence guarantees of AC algorithms. In fact, (...) it follows that in the limit both actors recover a policy that differs only by the level of entropy”. Furthermore, since SAC is an off-policy algorithm and DAC differs only by samples that land in the experience buffer, we believe it trivially follows that DAC retains convergence guarantees of SAC.
>
> > Currently the paper is written in a way that different updating rules are introduced; I expect to see a clear objective function (maybe with constraints) of Algorithm 1, so readers can easily see what it is optimizing, can the authors write it down in the rebuttal?
>
> The current version of the paper features a pseudo-code with links to every equation used in the calculation. Furthermore, Sections 3 & A2 detail what and why DAC is optimizing. If the Reviewer points us to what particularly the Reviewer finds confusing, we are happy to adjust the text.
>
> > When simultaneously maximizing both lower and upper bound of the Q values, would it squash all action values higher and still result in overestimation?
>
> We think that there is a misunderstanding. Overestimation is a property of the critic and is believed to stem from TD learning [2,5,6]. DAC features no “simultaneous maximization of lower and upper bound of the Q-values” and the critic used in DAC learns using a traditional clipped-double Q-learning [4,6]. The lower and upper bounds policies are optimized by conservative and optimistic actors respectively (not ‘simultaneously’), and the optimistic actor does not interact with TD learning in DAC design. As such, there is no reason for overestimation in DAC beyond overestimation that occurs in SAC. This is underlined in Section 1 of our manuscript: “DAC employs two actors, each independently optimized using gradient backpropagation with different objectives. The optimistic actor is trained to maximize an optimistic Q-value upper-bound while adjusting optimism levels automatically. This actor is responsible for exploration (sampling transitions added to the experience buffer). In contrast, the conservative actor is trained using standard lower-bound soft policy learning and is used for sampling temporal-difference (TD) targets and evaluation.”

---

> > ### Author Response · Authors · 2023-11-14
> > **Response to Reviewer SCgY (2)**
> >
> > > Verify the proposed method indeed mitigate overestimation comparing with an algorithm without using any correction, e.g., compare the estimate value and MC estimation;
> >
> > We added the proposed experiment on four DMC tasks of varying difficulty (500k steps, 5 seeds). There, we compare the overestimation of DAC, SAC and DAC using only optimistic actor. It shows that the decoupled design is indeed critical to mitigate bias bigger than that of standard SAC. We summarize the results in the table below.
> > |      Bias     | Conservative | Decoupled | Optimistic |
> > |:-------------:|:------------:|:---------:|:----------:|
> > |   hopper-hop  |      104     |     78    |     96     |
> > | quadruped-run |      638     |    442    |    2973    |
> > | humanoid-walk |     1112     |    1086   |    1671    |
> > |      mean     |      618     |    535    |    1580    |
> >
> > If the Reviewer wishes, we are happy to expand the above results to the high replay ratio regime.
> >
> > >  the current version of the paper directly using evaluation return as a performance measure, which, I think lacks justification, as it is unclear where the improved performance results from;
> >
> > To the best of our knowledge, evaluation return is the standard evaluation measure used in most of the RL literature. Additionally, Figure 4 presents evaluation of various DAC design decisions, which can be used to attribute performance gains to specific modules. Section F3 describes in detail why we were interested in evaluating those designs.
> >
> > > The added optimism is essentially an exploration strategy, some baselines aiming at exploration should be also compared.
> >
> > We evaluate against OAC and TOP (and SR versions thereof) which are baselines aiming at exploration via optimism. Additionally, we evaluate 10 simplifications of DAC which also feature optimism-driven exploration. Does the Reviewer have any specific optimism-based model-free continuous control baselines in mind?
> >
> > > Ablation study should be provided to justify the following design choices: the effect of ensemble (and do those baselines use ensemble too?)
> >
> > Using a critic ensemble has been standard in modern RL since TD3/SAC (which introduced clipped double Q-learning based on an ensemble of 2 critics) [1,2,5]. In Section 4 we write: “all algorithms except RedQ use the same network architectures and a standard ensemble of two critics”. As such, we believe that evaluating the effect of ensemble is outside of scope of this paper. If the Reviewer is curious about the effect of even bigger ensemble in continuous-control RL we kindly point to the following works [7,8]
> >
> > > the necessity of optimizing optimism and KL weight (can you use some intuitive choices instead of learning them)?
> >
> > Figure 4 of the original manuscript presents evaluation of various DAC design decisions, including not optimizing optimism and KL weight, optimizing only one of them and optimizing both. Section 3 discusses why it is hard to find “some intuitive choices instead of learning them” and Figure 3 shows an experiment backing that claim.
> >
> > > Any comments how do you decide the order of the updating rules 10-15?
> >
> > We added our additional modules (optimistic actor, optimism and KL) at the end of the update loop (after standard SAC updates). The optimistic actor update is dependent on optimism and KL. As such, we decided to update it ahead of the dual optimization modules. The order of optimism and KL is irrelevant as these modules are independent.
> >
> > > And how their learning rates are chosen?
> >
> > For the actor-critic modules, we use the same learning rates as the baselines proposed in earlier works. That includes the optimistic actor. For the optimism and KL modules, we took the same learning rate but divided it by 10 arbitrarily. We did not investigate the effects of varying learning rates.

---

> ### Author Response · Authors · 2023-11-14
> **Response to Reviewer SCgY (3)**
>
> > The algorithm introduces many more hyper parameters comparing with commonly seen SAC or TD3 due to the added components in the losses, I would not consider the comparison to be fair with other baselines, unless evidence of similar efforts have been made to thoroughly sweep baseline’s hyper-parameters is provided.
>
> We would like to respond to the above assertion in three points:
>
> 1. SAC/TD3 are arguably the most popular RL algorithms at the time of writing this response. Both were evaluated numerous times on DMC locomotion, which is one of the most popular RL continuous-control benchmarks. As such, we believe that the hyperparameter settings used for SAC/TD3 were arguably more “sweeped” than those of DAC. The hyperparameter choices used for our implementations of SAC and SR-SAC are used in the current DMC locomotion SOTA (SR-SAC) [9].
>
> 2. We kindly ask the Reviewer to note that reusing hyperparameters proposed in the earlier works is a common practice, especially in the case of such well-known benchmarks as DMC/gym [1,2,4,5,6,7,8,9,10,11].
>
> 3. Finally, we would like to point towards experiments presented in Figure 5 of our original manuscript. There, we evaluate 14 hyperparameter configurations of DAC (our entire sweep), as well as the reference SAC performance. According to the results, all 14/14 DAC configurations significantly outperform SAC.
>
> Considering the above arguments, we kindly ask the Reviewer to reconsider his “unfair comparison” assertion.
>
> > Eq 7, how do you calculate the gradient w.r.t optimistic actor parameters, it appears the gradient should be also propagated through/to the first two Q functions and these Qs are interdependent with actor.
>
> The Eq 7 is implemented the same way a standard backprob through critic is implemented in DDPG, D4PG, SAC, TD3, OAC, TOP etc. It is not required to update the critics parameters as a result of this update, we can just use the gradient calculation to propagate it further to the actor. If the Reviewer wishes, we can further expand on how it is usually implemented.
>
> > The motivation of the paper is to avoid overestimation while keep good exploration. Isn’t it quite intuitive to combine some exploration method with methods that mitigate overestimation? it shouldn’t be difficult to design such a baseline as mitigating overestimation typically require multiple critics and uncertainty estimate could be derived from the ensemble for exploration purpose. What is the proposed algorithm’s advantage?
>
> We evaluate against OAC, TOP, SAC, REDQ and TD3 which all “combine some exploration method with methods that mitigate overestimation” and all use multiple critics to derive an uncertainty estimate “for exploration purpose”. Furthermore, in Figure 4 we consider 10 DAC simplifications, which all as well fit the Reviewer’s description. We discuss the advantage of our approach in Section 3.
>
> > it shouldn’t be difficult to design such a baseline
>
> We design and evaluate a variety of baselines common to DAC. The results are presented in Figures 4 & 6. We are happy to add more baseline designs to our evaluations, if the Reviewer shares more details about the Reviewer’s envisioned baseline.
>
> > The experimental design of studying replay ratio appears to disconnect with the primary motivation of this paper; this should be put in the appendix.
>
> We argument why we study two replay ratio regimes in Section 1 & 4. In our view it is an important aspect in the light of modern RL studies, which was repeatedly shown to play a substantial role in the performance of RL algorithms. Also, as Reviewer XQ6P wanted us to expand the discussion of replay ratios, we would like to keep the analysis in the main body of the paper.
>
> We thank the Reviewer for their time and valuable input. In the light of new experimental results, we would like to ask the Reviewer to reconsider his score of the manuscript.

---

> > ### Author Response · Authors · 2023-11-14
> > **Response to Reviewer SCgY (4)**
> >
> > [1] Ciosek, Kamil, et al. "Better exploration with optimistic actor critic." Advances in Neural Information Processing Systems. 2019.
> >
> > [2] Moskovitz, Ted, et al. "Tactical optimism and pessimism for deep reinforcement learning." Advances in Neural Information Processing Systems. 2021.
> >
> > [3] Boyd, Stephen P., and Lieven Vandenberghe. Convex optimization. Cambridge university press. 2004.
> >
> > [4] Haarnoja, Tuomas, et al. "Soft actor-critic: Off-policy maximum entropy deep reinforcement learning with a stochastic actor." International Conference on Machine Learning. 2018.
> >
> > [5] Cetin, Edoardo, and Oya Celiktutan. "Learning pessimism for reinforcement learning." Proceedings of the AAAI Conference on Artificial Intelligence. 2023.
> >
> > [6] Fujimoto, Scott, Herke Hoof, and David Meger. "Addressing function approximation error in actor-critic methods." International Conference on Machine Learning. 2018.
> >
> > [7] Lee, Kimin, et al. "Sunrise: A simple unified framework for ensemble learning in deep reinforcement learning." International Conference on Machine Learning. 2021.
> >
> > [8] Chen, Xinyue, et al. "Randomized Ensembled Double Q-Learning: Learning Fast Without a Model." International Conference on Learning Representations. 2020.
> >
> > [9] D'Oro, Pierluca, et al. "Sample-Efficient Reinforcement Learning by Breaking the Replay Ratio Barrier." The Eleventh International Conference on Learning Representations. 2022.
> >
> > [10] Li, Qiyang, et al. "Efficient Deep Reinforcement Learning Requires Regulating Overfitting." The Eleventh International Conference on Learning Representations. 2022.
> >
> > [11] Ball, Philip J., et al. "Efficient online reinforcement learning with offline data." International Conference on Machine Learning. 2023.

---

> > ### Comment · Reviewer_SCgY · 2023-11-21
> >
> > Thank you for your efforts. I updated my score.

---

> > > ### Author Response · Authors · 2023-11-22
> > > **We thank the Reviewer for their suggestions**
> > >
> > > We thank the Reviewer for their input that helped us to improve the quality of our manuscript. We welcome any further suggestions and comments, in case the Reviewer has any.

---

### Author Response · Authors · 2023-11-14
**Meta-comment**

We thank the Reviewers for their time working on our paper. We are very happy that the following aspects of our work were acknowledged by the Reviewers: approach was described as ‘well-motivated’ [HF8k; 7ruN] and ‘solid' / 'very neat’ [HF8k; XQ6P]; the manuscript ‘well-written’ [XQ6P]; algorithm presentation ‘clear’ [SCgY]; performance gains ‘significant’ [HF8k] and ‘promising ’[XQ6P]; and that the number of experiments conducted was appreciated [HF8k, SCgY]. Reviewer feedback allowed us to make a number of improvements to the experimental section and the manuscript. Below we describe all implemented changes:

1. Harder tasks evaluation

Per suggestion of Reviewer XQ6P, we add additional evaluation of dog (walk, run, trot) and humanoid (walk, run) domains on 1mln and 3mln environment steps. There, we evaluate SR-SAC (current model-free SOTA), DAC (approach proposed in our manuscript) and TD-MPC [1] (model-based SOTA suggested by Reviewer XQ6P). We describe the results in the table below. **To the best of our knowledge, the results indicate that DAC achieves the highest recorded sample efficiency of a RL algorithm on the dog and humanoid domains.**

|    task   | SR-SAC |   DAC   |   DAC+  | TD-MPC | DAC+ | TD-MPC |
|:---------:|:------:|:-------:|:-------:|:------:|:----:|:------:|
|  hum-walk |   723  | **852** |   846   |   727  |  **918** |   912  |
|  hum-run  |   203  |   228   | **243** |   226  |  **397** |   388  |
|  dog-walk |   56   |   552   | **910** |   862  |  **953** |   939  |
|  dog-trot |   37   |   361   | **635** |   591  |  868 |   **886**  |
|  dog-run  |   51   |   204   | **402** |   323  |  **642** |   502  |
|    mean   |   214  |   439   | **607** |   546  |  **756** |   725  |
| env steps |  1mln  |   1mln  |   1mln  |  1mln  | 3mln |  3mln  |

We will add the results for 3mln steps when the training finishes (we are also happy to share wandb training logs as well). We note that DAC in standard configuration achieves substantially better results than SR-SAC and is close to model-based SOTA that uses more compute and bigger networks. When we add a single hidden layer and increase the copying frequency (denoted as DAC+), DAC exceeds the performance of model-based TD-MPC on 1mln environment steps. Furthermore, in contrast to DAC, SR-SAC appears to not solve dog tasks at all (a result which is consistent with [1]).

2. Overestimation measurement

Per suggestion of Reviewer SCgY, we add an experiment that verifies that the decoupled architecture mitigates overestimation stemming from the optimistic policy. We calculate overestimation by comparing evaluation returns with entropy and the critic output. The table below summarizes the average absolute bias recorded during the experiment.

|      Bias     | Conservative | Decoupled | Optimistic |
|:-------------:|:------------:|:---------:|:----------:|
|   hopper-hop  |      104     |     78    |     96     |
| quadruped-run |      638     |    442    |    2973    |
| humanoid-walk |     1112     |    1086   |    1671    |
|      mean     |      618     |    535    |    1580    |

Interestingly, we observe that the decoupled architecture yields not only comparable, but slightly smaller bias than the baseline conservative architecture (SAC). We hypothesize that this might be connected to the better state-action coverage of DAC as compared to SAC discussed in Section 3. Naturally, optimistic configuration yields most bias, especially in very dense reward environments.

3. New baselines

We add TD-MPC (DMC locomotion model-based SOTA), D4PG and MPO (per suggestion of Reviewer XQ6P).

4. Changes to the manuscript

We decided to apply a number of changes to our manuscript. Those changes will appear in a revised version of the paper which we will upload in the upcoming days - the changes will be coloured red for easier read. The changes include:

a. An Appendix section describing the results of 3mln step evaluation against SR-SAC and TD-MPC suggested by Reviewer XQ6P

b. An Appendix section describing the results of overestimation evaluation suggested by Reviewer SCgY

c. Additional description for abbreviations and Figures 1 & 2 suggested by Reviewer HF8k

d. Explicit statement about limitations that DAC inherits from SAC suggested by Reviewer HF8k

e. Discussion of additional Related Work suggested by Reviewer XQ6P

f. Minor typos and wording changes.

[1] Hansen, N., X. Wang, and H. Su. "Temporal Difference Learning for Model Predictive Control." International Conference on Machine Learning, PMLR. 2022

---

> ### Author Response · Authors · 2023-11-17
> **Meta-comment 2**
>
> We thank the Reviewers again for their time and suggestions on how to improve the quality of our manuscript. Following our commitment, we uploaded a revised version of the manuscript. In the revised version, we colour major changes red for an easy read. We summarize the implemented changes below.
>
> *ADDITIONAL EXPERIMENTS* - we add a paragraph describing the experiments on the dog domain, as well as extended runs on the humanoid domain. We add the two evaluations on this domain:
> 1. 1mln environment steps - where we run the algorithms in a setting consistent with the high replay ratio regime
> 2. 3mln environment steps - where we make adjustments to DAC such that it is comparable to the model-based approach that yields state-of-the-art performance [1]
>
> We detail results for the hard DMC tasks suggested by the Reviewer XQ6P. **To the best of our knowledge, DAC achieves the highest recorded sample efficiency and final performance achieved by an RL algorithm on the “Run” tasks of dog and humanoid domains**. The results are presented in Section 4 and Appendix D.3 of the revised manuscript. We summarize the results in the table below.
>
> |   step   | SR-SAC |   DAC   | TD-MPC |   DAC+  |
> |:--------:|:------:|:------:|:-------:|:-------:|
> |   1mln   |   214  |   439  |   546   |   607   |
> |   3mln   |   NA   |   NA   |   725   |   756   |
>
> *OVERESTIMATION TESTS* - we added an Appendix section describing the overestimation experiments suggested by the Reviewer SCgY. There we evaluate Q-value overestimation implied by a critic updated in three regimes:
>
> 1. Conservative - exploration and TD learning are both conservative
> 2. Optimistic - exploration and TD learning are both optimistic
> 3. Decoupled - optimistic exploration and conservative TD learning
>
> We find that the decoupled architecture yields the least overestimation from the compared methods. We hypothesize that the slightly lower overestimation than the conservative architecture stems from better state-action coverage of the decoupled architecture which we discuss in Section 3. We describe the results in Appendix D.1 of the revised manuscript.
>
> *NEW BASELINES* - we added new baseline algorithms suggested by the Reviewer XQ6P: MPO; D4PG and TD-MPC. The comparison is presented in Figure 1 and Appendix H of the revised manuscript.
>
> *OTHER MINOR CHANGES* - we expanded related work; added new information to the hyperparameters and experiments descriptions; added information on the newly added baseline algorithms; added a section on replay ratio and RL; added some minor changes, fixed some typos and missing words.
>
> We hope that the newly added content reinforces the confidence in our proposed method and answers the weaknesses discussed by the Reviewers. If any of the Reviewer has any further suggestions, we are happy to implement them.
>
> [1] Hansen, N., X. Wang, and H. Su. "Temporal Difference Learning for Model Predictive Control." International Conference on Machine Learning, PMLR. 2022

---

### Comment · Area_Chair_2QEC · 2023-11-20

Dear Reviewers,

The author reviewer discussion period is ending soon this Wed. The current evaluation is quite borderline and the authors have provided extensive responses including new empirical results. It would be much appreciated if you all could check the author response to see if your concerns are cleared or there are still outstanding items that you would like to have more discussion.

Thanks again for your service to the community.

Best,
AC

---

### Meta-Review · Area_Chair_2QEC · 2023-12-05

**Metareview:**

This work proposes to use two actors to achieve optimism and conservation separately. Performance improvements are observed in empirical study.

Strengths: The experiments are extensive and improvements are observed over baselines.

Weakness: This paper combines many different ideas / ingredients from many existing works. Despite that this combination appears novel, none of the individual idea seems new. Moreover, there is not a theoretical analysis that oversees the entire architecture, analyzing how each component affects the other and where the entire ship goes.

It is also worth mentioning that in Figure 1 (a), the OAC and TD3 perform exactly the same, which might be from some bug of the plotting program. I remark that this is just a side note and does not factor into the decision at all.

**Justification For Why Not Higher Score:**

Lack of theoretical analysis on the interaction between components.

**Justification For Why Not Lower Score:**

N/A

---

### Decision · Program_Chairs · 2024-01-16

Reject